# Negative refraction of light in an atomic medium

L. Ruks[1,2,3] ✉, K. E. Ballantine[4] & J. Ruostekoski [4] ✉

The quest to manipulate light propagation in ways not possible with natural media has driven the development of artificially structured metamaterials. One of the most striking effects is negative refraction, where the light beam deflects away from the boundary normal. However, due to material characteristics, the applications of this phenomenon, such as lensing that surpasses the diffraction limit, have been constrained. Here, we demonstrate negative refraction of light in an atomic medium without the use of artificial metamaterials, employing essentially exact simulations of light propagation. High transmission negative refraction is achieved in atomic arrays for different level structures and lattice constants, within the scope of currently realised experimental systems. We introduce an intuitive description of negative refraction based on collective excitation bands, whose transverse group velocities are antiparallel to the excitation quasi-momenta. We also illustrate how this phenomenon is robust to lattice imperfections and can be significantly enhanced through subradiance.

Negative refraction of electromagnetic waves[1–7] is characterised by counter-intuitive phenomena, such as the bending of waves in a medium in the opposite direction to what normally occurs and the amplification of evanescent waves in the bulk[8,9]. These entail possibilities of transformative applications, including cloaking[10–12] and superlensing[9,13], and have also gained considerable attention in other wave types, such as elastic and acoustic waves[14,15]. Metamaterials, synthetic man-made media, were developed to overcome limitations of natural materials, with these unconventional refraction phenomena serving as a key driving force behind their advancement. Despite the significant interest in negative refraction, intrinsic non-radiative damping and ever-present fabrication imperfections of resonators in metamaterials result in significant losses at optical frequencies[16,17]. Similarly, negative refraction in 3D photonic crystals[18,19] is constrained by the refractive indices of their constituent dielectrics, is sensitive to imperfections, and operates within a narrow range of incidence angles with limited subwavelength resolution due to the requirement of wavelength-scale periodicity. Consequently, a practical application of negative refraction of light has yet to be demonstrated.

Here we show that negative refraction of light can be obtained in atomic media without the use of artificially fabricated resonators. This is made possible by harnessing and utilising cooperative, non-local atom responses that arise from strong light-mediated interactions at high densities. We achieve this controlled response by considering subwavelength atomic arrays[20] with unit filling, which can be experimentally realised in system sizes up to tens of wavelengths[21]. Cooperative optical interactions in atomic arrays have recently been confirmed in transmitted light, as evidenced by the spectral resonance narrowing below the fundamental quantum limit dictated by a single atom[22] and coherently manipulated optical switching[23]. In contrast to our approach, previous proposals for utilising atomic media to achieve negative refraction have relied on quantum interference effects in multilevel systems involving inherently weak magnetic dipole transitions for independently scattering atoms[24,25]. These approaches necessitate, e.g., exceptionally high refractive indices at high densities, that currently surpass the limits of experimental capabilities[26].

We demonstrate the negative refraction of light through microscopic, atom-by-atom simulations that exactly incorporate all

[1]NTT Basic Research Laboratories, NTT Corporation, 3-1 Morinosato Wakamiya, Atsugi, Kanagawa 243-0198, Japan. [2]NTT Research Center for Theoretical Quantum Information, NTT Corporation, 3-1 Morinosato Wakamiya, Atsugi, Kanagawa 243-0198, Japan. [3]Quantum Systems Unit, Okinawa Institute of Science and Technology Graduate University, Onna-son Okinawa 904-0495, Japan. [4]Department of Physics, Lancaster University, Lancaster LA1 4YB, UK. ✉e-mail: lewis.ruks@ntt.com; j.ruostekoski@lancaster.ac.uk

recurrent scattering processes between stationary atoms in the low light intensity limit[27,28], and accurately describe experimental conditions[22,29]. Our analysis encompasses both the $J = 0 \rightarrow J' = 1$ transition, typical of experimentally relevant alkaline-earth-metals and rare-earth metals (e.g., Sr, Yb), and a two-level transition found in cycling transitions for alkali-metal atoms, using the optical lattice spacing of Rb from light transmission experiments[22,23]. The simulations uncover a distinct deflection of propagating light beams in the opposite direction to what normally occurs. This enables us to extract from microscopic simulations the negative value of refractive index. We demonstrate high-transmission negative refraction that is resilient to lattice imperfections, such as missing atoms and position uncertainty, over a range of laser frequencies, incident angles, and lattice constants by modelling 3D atom arrays as stacked infinite 2D planar layers when in each layer the scattering problem is represented in momentum space. Our findings, confirmed for finite-sized arrays in a few-layer scenario through large-scale simulations, reveal that such refraction aligns with the simple phenomenology of transverse Bloch band collective resonances. Notably, negative refraction can be realised by engineering and exciting such resonance bands whose in-plane group velocities are antiparallel to the excitation quasimomenta—which are a common occurrence in moderately subwavelength arrays. Intriguingly, the effective refractive index can be significantly enhanced through coupling with more subradiant excitations.

## Results

We consider a 3D cubic Bravais lattice of $N_x \times N_y \times N_z$ atoms, primarily examining the large array limit within the $yz$ plane. To facilitate this limit, we represent the system as stacked planar square arrays with a lattice constant $a$, positioned at planes $x = x_\ell = \ell a$ ($\ell = 0, ..., N_x - 1$), as illustrated in Fig. 1a. We assume the low light intensity limit where a coherent monochromatic Gaussian laser, with the amplitude $\mathcal{E}^+(\mathbf{r})$, dominant wavevector $\mathbf{k} = (k_x, \mathbf{k}_\parallel)$, and wavelength $\lambda = 2\pi/k$, is incident from $x = -\infty$ propagating in the positive $x$ direction at an angle $\theta$ to the $y$-axis, such that $k_z = 0$. Unless the lattice is finite, we further assume the beam is confined in the $xy$ plane only (Supplementary Note 1). For the $J = 0 \rightarrow J' = 1$ atomic transition, we write the dipole excitation as a vector $\mathcal{D}\mathcal{P}^{(\mathbf{n})}$ in atom $\mathbf{n}$, where $\mathcal{D}$ denotes the reduced dipole matrix element and $\mathbf{n} = (\ell, j)$ indexes the atom at position $\mathbf{r_n} = \ell a\hat{\mathbf{x}} + \mathbf{r}_{\parallel j}$ for the in-plane lattice vector $\mathbf{r}_{\parallel j}$. For simulation cases that consider a two-level transition with unit polarisation vector $\hat{\mathbf{e}}_\nu$, the atomic dipole amplitude, $\mathcal{P}^{(\mathbf{n})} = \hat{\mathbf{e}}_\nu \mathcal{P}^{(\mathbf{n})}$, is reduced to a scalar. We denote the light and atomic excitation amplitudes as slowly varying positive-frequency components, with rapid oscillations $e^{-i\omega t}$ at the laser frequency $\omega$ factored out[30]. The atomic dipole amplitude of the atom $\mathbf{n}$ is driven by the incident field Rabi frequency $\mathcal{R}^+(\mathbf{r_n}) = \mathcal{D}\mathcal{E}^+(\mathbf{r_n})/\hbar$ and the field scattered by all the other atoms, and satisfies the

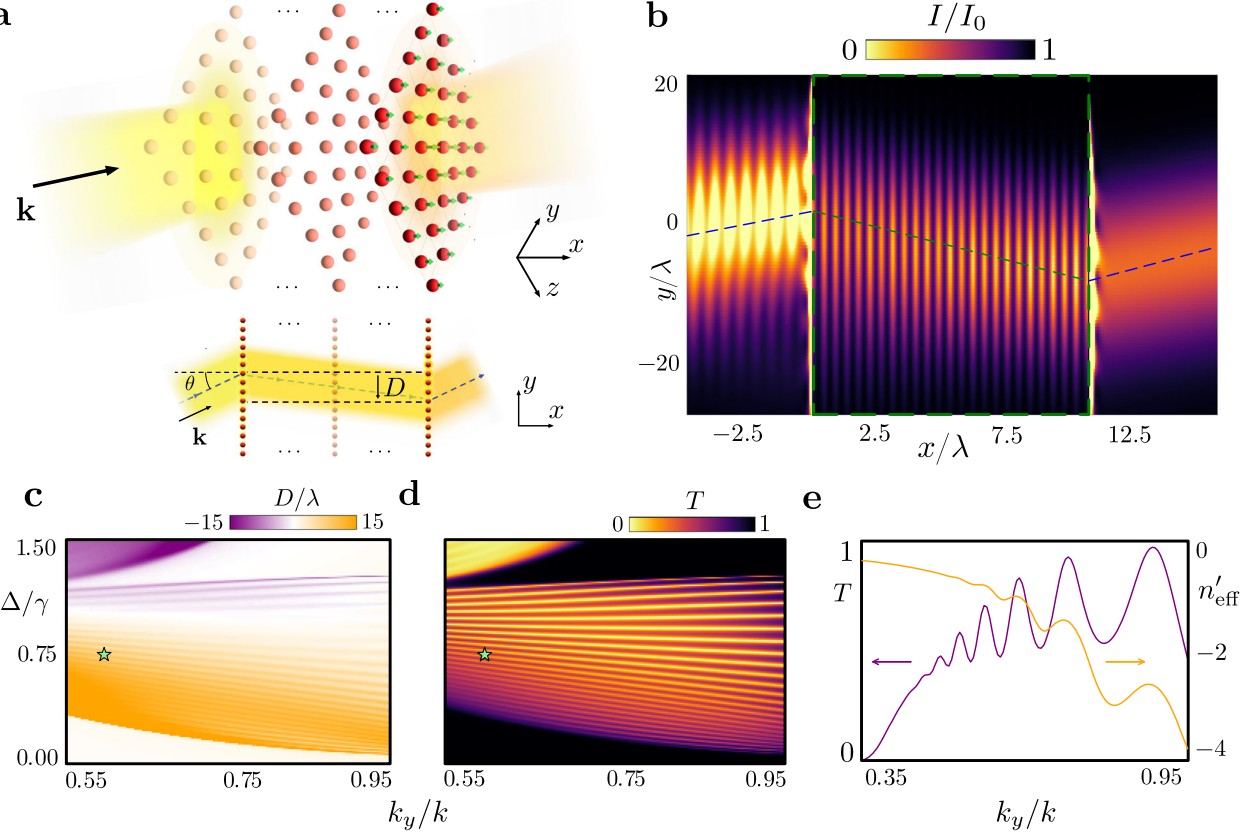

**Fig. 1 | Negative refraction and transmission of light through an atomic medium. a** Schematic shows light transmission through a cubic atom array, stacked in infinite planar lattices along the $x$-axis, with the incident beam propagating towards $x = \infty$. The dominant wavevector $\mathbf{k}$ lies in the $xy$ plane at angle $\theta = \arcsin(k_y/k)$ to the lattice normal, tilting in the $y$ direction. The transmitted beam undergoes lateral displacement $D$ along the $y$-axis. **b** Negative refraction of a beam incident with $\theta = 0.2 \times \pi$ and laser detuning $\Delta/\gamma = 0.73$ from the atomic resonance, in units of single-atom linewidth $\gamma$, through a 25-layer atomic lattice for the $J = 0 \rightarrow J' = 1$ transition. This is manifested in the normalised light intensity profile $I/I_0$ (outside the medium) and atomic polarisation density $|\langle\hat{\mathbf{P}}^+\rangle|^2/|\langle\hat{\mathbf{P}}^+\rangle|^2_{\max}$ (within the lattice delimited by the dashed green box) at plane $z = 0$, scaled by resonance wavelength $\lambda$, where $I_0 = 2\epsilon_0 c \max_{\mathbf{r}} |\mathcal{E}^+(\mathbf{r})|^2$ represents the maximum incident intensity. For visualisation, $|\langle\hat{\mathbf{P}}^+\rangle|^2/|\langle\hat{\mathbf{P}}^+\rangle|^2_{\max}$ for point-like atoms is smoothed by convolution with a Gaussian of the root-mean-square widths $\sigma_x = 0.25a$ and $\sigma_y = 0.5a$. The blue dashed lines trace the peak light intensity, while the connecting green line marks the effective trajectory in the medium. **c** $D/\lambda$ and **d** power transmission $T$ as a function of $\Delta/\gamma$ and $k_y/k$ across the transmission band. Green stars denote the parameters taken in **b** for the incident beam. **e** Variation of $T$ and effective group refractive index $n'_{\mathrm{eff}}$ with $k_y/k$, for the same laser detuning as in (**b**).

differential equation of coupled driven linear oscillators

$$\dot{\mathcal{P}}^{(\mathbf{n})} = (i\Delta - \gamma)\mathcal{P}^{(\mathbf{n})} + i\mathcal{R}^+(\mathbf{r_n}) + i\xi \sum_{\mathbf{n}' \neq \mathbf{n}} \mathrm{G}(\mathbf{r_n} - \mathbf{r_{n'}})\mathcal{P}^{(\mathbf{n}')}, \quad (1)$$

where $\mathrm{G}(\mathbf{r})$ is the free-space dipole radiation kernel[31], $\xi = \mathcal{D}^2/(\hbar\epsilon_0)$, $\gamma$ the single-atom linewidth, and $\Delta = \omega - \omega_0$ the laser detuning from the atomic resonance frequency $\omega_0$. More details of the formalism are provided in Supplementary Note 1.

While the linear system Eq. (1) can be solved in the steady state to obtain the radiative excitations and to determine the total coherently scattered field $\epsilon_0\langle\hat{\mathbf{E}}_s^+(\mathbf{r})\rangle = \sum_{\mathbf{n}}\mathrm{G}(\mathbf{r} - \mathbf{r_n})\mathcal{D}\mathcal{P}^{(\mathbf{n})}$, this becomes computationally prohibitive in large systems. To calculate the transmission and refraction of light through large atomic ensembles, we instead utilise a momentum space representation of Eq. (1) (Methods). This involves considering sufficiently large atomic layers that can be approximated as translationally invariant along the $y$- and $z$-axes, with well-defined excitation quasimomenta. This approach enables us to construct effective light propagators for each atomic layer in the $yz$ plane through a numerically efficient momentum-space summation of radiative interactions within those layers. The momentum-space representation naturally includes the high-frequency cutoff implicit in nonrelativistic electrodynamics[30], and permits regularisation of the high-momentum divergence, while avoiding mathematical issues[32] stemming from the lack of absolute convergence in these sums (Supplementary Note 1). The atomic polarisation amplitudes of Eq. (1) for the $\ell$th layer with in-plane Bloch wavevector $\mathbf{q}_{\parallel}$, determined by the propagating light, are represented by Bloch waves $\mathcal{P}^{(\ell,j)} = \sum_{\mathbf{q}_{\parallel}} \mathcal{P}_\ell(\mathbf{q}_{\parallel})e^{i\mathbf{q}_{\parallel}\cdot\mathbf{r}_{lj}}$ (Methods). The dynamics of coupled layers then resembles that of 1D electrodynamics in waveguides[33] where each layer forms an effective superatom[34–36]. However, as a crucial deviation from the ideal 1D propagation, coupling between in-plane and out-of-plane polarisation components are present, along with decaying evanescent (near-field) contributions due to higher Bragg scattering orders. This approach allows for efficient numerical computation of radiative excitations and the scattered light.

Given that $k_z = 0$ and due to the symmetry in an infinite layer, the beam selectively excites only Fourier components $\mathcal{P}_\ell(q_y, q_z = 0)$ along the principal lattice axis. As a result, the contribution from the $|J = 0, m = 0\rangle \rightarrow |J' = 1, m = 0\rangle$ transition (for quantisation axis along $z$) cancels out in the optical response for the studied $p$-polarised incident light. Negative refraction is not observed in the current atomic array for $s$-polarised light. However, the possibility for negative refraction with the $s$-polarisation is not in general precluded, as strong cooperative magnetic dipole responses can be induced through more specifically tailored atomic geometries[37,38].

In Fig. 1b, we demonstrate cooperative negative refraction of light through a 25-infinite-layer atomic medium ($N_x = 25$, $N_y = N_z = \infty$) with a $J = 0 \rightarrow J' = 1$ transition and the lattice constant $a = 0.45\lambda$. This illustration is based on atom-by-atom simulations for the propagating beam incident at angle $\theta = \arcsin(k_y/k) = 0.2 \times \pi$ relative to the array normal. The negative refraction of light is revealed by a negative $y$-displacement $D \simeq -9\lambda$ (see Fig. 1a) from $y = 0$ in the coherent transmitted light intensity profile $I = 2\epsilon_0 c|\langle\hat{\mathbf{E}}^+(\mathbf{r})\rangle|^2$, for the total coherent field $\langle\hat{\mathbf{E}}^+(\mathbf{r})\rangle = \mathcal{E}^+(\mathbf{r}) + \langle\hat{\mathbf{E}}_s^+(\mathbf{r})\rangle$, whilst the deflection of the beam within the medium is clearly manifested in the accompanying atomic polarisation profile. The propagation direction and displacement of the coherent beam unambiguously determine refraction-like[19] behaviour due to subwavelength periodicity of the lattice that results in all non-zero diffraction orders of the transmitted beam to be evanescent. We find a high power transmission $T \simeq 0.8$ despite the large medium thickness $(N_x - 1)a = 10.8 \times \lambda$ of many wavelengths as, in sharp contrast to metamaterials, losses in atomic ensembles for fixed positions arise solely from the scattering at finite in-plane boundaries[39].

In Fig. 1c–e, we show the beam displacement $D$, measured in the positive $y$-direction, from the maximum intensity at the entrance face

to that at the exit face, and power transmission $T$ across various light detunings and incident angles. Collective resonances, forming a transmission band, are characterised by local maxima in $T$ and $|D|$. In the chosen system, negative refraction ($D < 0$)—such as the case shown in Fig. 1b—is prevalent. Remarkably, transmission resonance maxima approach unity, with deviations from perfect transmission due to the finite beam waist. Well within the transmission band, we generally observe high transmission taking a minimum of about $T \simeq 0.5$, with minima approaching zero towards the edge of the transmission band. The oscillations near unit transmission within a broad background $T \simeq 0$ are characteristic of Fano resonances, arising from coupling between superradiant and subradiant excitations formed by scattering between layers, analogously to behaviour in chains of atoms coupled through 1D waveguides[33]. For each incident angle, both the number of resonances and inverse linewidths of resonance within the transmission band then grow linearly with the number of layers. The beam displacement, which has no analogue in 1D electrodynamics, exhibits oscillations in phase with transmission and is also found to grow linearly with medium thickness inside the transmission band, as the travel distance of the deflected beam is extended with increasing layer number.

Negative refraction of light, shown in Fig. 1, occurs for the $J = 0 \rightarrow J' = 1$ transition found in alkaline-earth-metals and rare-earth metals such as Sr and Yb, where the Mott-insulator transition in optical lattices has been observed[40,41]. In contrast, the experiments on light transmission through Mott-insulator states in optical lattices in refs. 22,23 use Rb, with the lattice constant $a \simeq 0.68\lambda$. Similar to other alkaline-metal atoms, Rb can utilise an isolated two-level cycling transition. In Fig. 2, we show transmission of light through a cubic array of two-level atoms with the same lattice constant, consisting of five infinite planar arrays. Despite the absence of isotropic atomic polarisation and the significantly thinner sample, the negative refraction of light is distinctly observable. We find a lateral displacement $D \simeq -\lambda$ for a beam with reduced waist $w = 5\lambda$, ($1/e$ radius of the field amplitude) incident at an oblique angle $\theta = \arcsin(0.2)$, achieving high transmission $T \simeq 0.95$.

The propagation of light shown in Figs. 1 and 2 is based on precise microscopic atomic simulations. The transition from microscopic descriptions to the macroscopic electrodynamics of continuous media —characterised by bulk material parameters—is highly non-trivial. Indeed, exact simulations have dramatically shown that standard continuous media electrodynamics, which is based on the coarse-

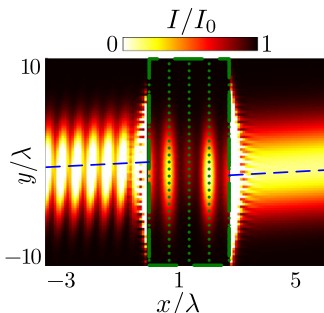

**Fig. 2 | Negative refraction in a 5-layer lattice of two-level atoms with Rb lattice spacing.** Transmission of light through a cubic array of atoms, formed by five stacked infinite planar lattices. Each atom, denoted by a green dot, exhibits an isolated $\sigma^+$-polarised two-level transition, with the quantisation axis along $z$, and a lattice constant $a = 0.68\lambda$, corresponding to Mott insulator-state experiments of Rb atoms[22,23]. Negative refraction is demonstrated by normalised light intensity $I/I_0$ outside the medium at the plane $z = 0$, where $I_0 = 2\epsilon_0 c \max_\mathbf{r}|\mathcal{E}^+(\mathbf{r})|^2$ represents the maximum incident field intensity. Within the lattice, delimited by the dashed green box, the atomic polarisation density $|\langle\hat{\mathbf{P}}^+\rangle|^2/|\langle\hat{\mathbf{P}}^+\rangle|^2_{max}$ is visualised and smoothed using convolution with a Gaussian of root-mean-square widths $\sigma_x = 0.25$, $\sigma_y = 0.5$. The light beam, incident in the $xy$ plane at angle $\theta = \arcsin(0.2)$ to the lattice normal and with a detuning $\Delta = -0.1\gamma$ from the atomic resonance, shows a lateral displacement along the $y$-axis of approximately $-\lambda$ and power transmission $T \simeq 0.95$.

grained averaging over precise atomic positions in granular samples, can be significantly violated[42–45]. In this context, phase velocities for Bloch waves excited in discrete atomic layers, or a refractive index for the medium, generally cannot be defined in the conventional sense. To determine the effective group refractive index and quantify the beam's deflection, we define the real part of the refractive index[46], $n'_{eff}$, for each angle of incidence by analogy with the Snell-Descartes law, $\sin\theta = n'_{eff}\sin\theta'$, where the effective deflection angle $\theta' = \arctan(D/[(N_x - 1)a])$ is calculated using the beam displacement and medium thickness $(N_x - 1)a$. In Fig. 1b, we find $n'_{eff} \simeq -0.5$.

Considering a fixed detuning $\Delta = 0.73\gamma$ in Fig. 1c, d, we have confirmed that negative refraction, with $n'_{eff} \lesssim -0.5$, persists over a broad range of incidence angles $\pi/6 \lesssim \theta \lesssim \pi/2$ (Fig. 1e), whilst $n'_{eff} < 0$ more generally is found in the region, $0 \lesssim \Delta \lesssim \gamma$, of $D < 0$ occupying a large fraction of the transmission band in Fig. 1c. Oscillatory behaviour in the negative effective group index as a function of the incident angle aligns with similar patterns found in the lateral displacement in Fig. 1c. Standard positive refraction is observed in the narrow upper region in Fig. 1c, with a specific instance demonstrated in Supplementary Fig. 1.

In contrast to metamaterials, which require material optimisation of individual elements, negative refraction in atomic systems can be engineered solely from the strong collective response arising from the lattice geometry. The atomic configurations are not subject to stringent conditions, and we have generally observed negative refraction when $a \lesssim \lambda$. It is important to note that equal in-plane and out-of-plane lattice spacings are not a prerequisite for observing negative refraction. Our formalism readily generalises to different spacings, where we have also observed qualitatively similar results, even when the lattice constant in one dimension exceeds the resonance wavelength.

To intuitively elucidate the optical response of finitely many stacks, we cast the multiple scattering dynamics into the form of an eigenvalue problem (see Methods). The eigenmodes represent collective resonances of atomic polarisation profiles, each with a well-defined in-plane quasimomentum $\mathbf{q}_{\parallel}$. The accompanying eigenvalues define resonance linewidths $v^{(j)}(\mathbf{q}_{\parallel})$ and line shifts $\delta^{(j)}(\mathbf{q}_{\parallel})$, observed in Fig. 3a for $N_x = 25$ layers infinite in the $xy$ plane and the $J = 0 \rightarrow J' = 1$ transition, for each band indexed by $j = 1...,3N_x$. Whilst the number of bands is determined by the layer number and the number of transitions, comparison with Fig. 3b reveals how the geometry of the bands emerges from the otherwise identical few-layer medium, with $N_x = 5$, which already exhibit qualitatively similar line shift profiles. The in-plane band structure is commonly employed to determine transmission also through single layer atomic arrays[20,36,47]. By diagonalising a $3N_x \times 3N_x$ matrix for each $\mathbf{q}_{\parallel}$, we efficiently characterise transmission through $N_x$ layers, facilitating numerically feasible calculations for large finite 3D structures[46] whose optical responses are not accurately predicted by the fully-infinite 3D band structure[48].

In Fig. 3c, d, we present the displacement of a beam resonant at each of the collective excitation bands in the few layer ($N_x = 5$) and many-layer ($N_x = 25$) cases, respectively. Refraction is then understood in terms of collective polarisation eigenmodes excited by the incident field, with an illustration given in Fig. 4a. A spectrally narrow beam excites a coherent polarisation wavepacket with in-plane wavevectors centred around $\mathbf{k}_{\parallel}$, at resonance $\Delta \simeq -\delta^{(j)}(\mathbf{q}_{\parallel})$ for band $j$. The group velocity of a wavepacket is determined by the gradient of the dispersion band, $\mathbf{v}_g = -\nabla_{\mathbf{q}}\delta^{(j)}(\mathbf{q})$, where the derivative is taken with respect to the quasimomentum $\mathbf{q}$. The light-induced excitation wavepacket experiences transverse displacement along the $y$ axis due to propagation at the group velocity component in that direction, $v_{g,y} = -\partial\delta^{(j)}(\mathbf{q}_{\parallel})/\partial q_y$. The lifetime of this excitation is given by the inverse of the collective linewidth of the excited eigenmode $1/v^{(j)}(\mathbf{q}_{\parallel})$. We can approximate $-\partial\delta^{(j)}(\mathbf{q}_{\parallel})/\partial q_y$ and $v^{(j)}(\mathbf{q}_{\parallel})$, for all appreciably excited quasimomenta $\mathbf{q}_{\parallel}$, by their values at $\mathbf{q}_{\parallel} = \mathbf{k}_{\parallel}$, after neglecting terms of the next order in the inverse beam waist. We then find that the beam accumulates a transverse displacement

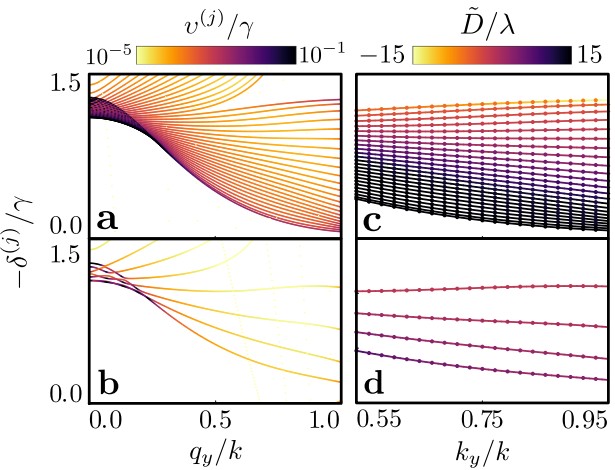

**Fig. 3 | Collective resonance excitation band structure and beam displacement for the $J = 0 \rightarrow J' = 1$ transition. a** 25-layer **b** 5-layer collective line shifts $\delta^{(j)}(q_y, q_z = 0)$, in units of the single-atom linewidth $\gamma$, for atomic Bloch wave resonances across bands indexed by $j$, as a function of in-plane quasimomentum $q_y$, indicative of the incident light's tilting angle. The lattice spacing $a = 0.45\lambda$, in units of resonance wavelength $\lambda$. The colour coding indicates the collective resonance linewidth (see Methods), $v^{(j)}(q_y, q_z = 0)$, on a logarithmic scale, normalised to $\gamma$. **c** 25-layer **d** 5-layer exact (scatter points) lateral displacement $D(\mathbf{k}_{\parallel}, -\delta^{(j)}(k_y, 0))$, in units of $\lambda$, compared with approximate (solid lines) lateral beam displacement $\tilde{D}(\mathbf{k}_{\parallel}, -\delta^{(j)}(k_y, 0)) = v_{g,y}^{(j)}/v^{(j)}(k_y, 0)$, derived from the group velocity $v_{g,y}^{(j)} = -\partial\delta^{(j)}(k_y, 0)/\partial k_y$ along the $y$-axis for laser detuning $\Delta = -\delta^{(j)}(k_y, 0)$ resonant with band $j$, considering the incident light's wavevector $y$-component $k_y$.

$D \simeq \tilde{D}(\mathbf{k}_{\parallel}, \Delta) = -\partial\delta^{(j)}(\mathbf{k}_{\parallel})/\partial q_y \times 1/v^{(j)}(\mathbf{k}_{\parallel})$ over the collective mode lifetime. The group velocity thus determines, through $\tilde{D}$, the sign of the effective group refractive index. Comparing $\tilde{D}$ with the exact displacement across the transmission band (Fig. 3c), we find the basic approximation $\tilde{D} \simeq D$ to be remarkably accurate at resonance. Intriguingly, narrow resonance linewidths result in dramatically increased displacements, indicating a strong enhancement of effective refraction due to subradiant collective radiative excitation eigenmodes. This effect is observed irrespective of the effective group-index sign, whilst the approximation continues to hold for few-layer media (Fig. 3d).

Our analysis of the lateral displacement also allows us to derive scaling behaviour of the optical response as a function of the sample thickness, achieving the macroscopic 'bulk' behaviour limit of light refraction. The emergence of the refractive index from the microscopic principles is illustrated in Fig. 4b–d where the number of layers is varied from 25 to 100, effectively maintaining the macroscopic refractive response while the collective transverse dispersion band density increases. This scaling is possible because the collective linewidth $v \propto 1/N_x$ precisely compensates for the increasing layer number $N_x$, such that $(N_x - 1)v$ remains finite for large $N_x$. The finite limit of $(N_x - 1)v$ for phase-matched resonances within the transmission band is consistent with the observed $1/N_x$ linewidth scaling of light-coupled resonant excitations in atomic arrays[34], and directly demonstrates the linear dependence of $D$ on thickness in macroscopic media.

Our examination of negative refraction, employing stacked infinite layers, offers a computationally powerful approach even for large systems. However, in realistic finite arrays, edge effects may alter the excitations and lead to scattering off the sample boundaries. Consequently, we directly calculate the transmission of light, using Eq. (1), for a finite lattice of $N_x \times N_y \times N_z = 5 \times 25 \times 25$ two-level atoms in Fig. 5a. When compared with the case of infinite in-plane arrays (Fig. 2) we observe an excellent match in the exact scattering profile for $z = 0$, which exhibits a negative refraction predicted by the resonant polarisation Bloch bands (Fig. 5b) at $q_z = 0$. These results are consistent with experimental observations of light transmission through an atomic

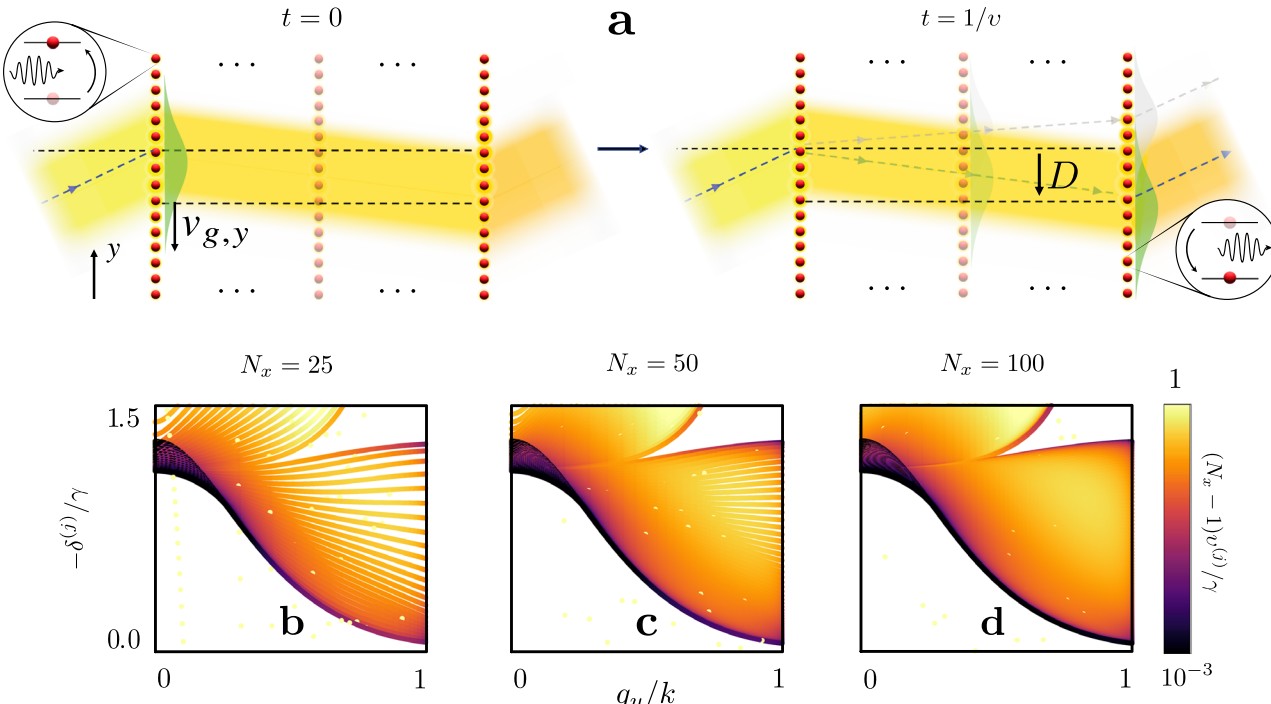

**Fig. 4 | Microscopic origin of negative refraction and emergence of macroscopic optical bulk response. a** Schematic illustrates the microscopic mechanism of negative refraction. An atomic polarisation wavepacket, excited by the incident beam, propagates along the $y$-axis over its lifetime $t = 1/v$, where $v = v^{(j)}(q_y, q_z = 0)$ is the collective linewidth, accumulating a transverse displacement of $D \simeq v_{g,y} \times 1/v$. The transverse group velocity component $v_{g,y} = -\partial \delta^{(j)}(q_y, q_z = 0)/\partial q_y$ is derived from the collective line shifts $\delta^{(j)}(q_y, q_z = 0)$ for phase-matched quasimomenta $q_y$ in resonant band $j$. Green and grey wavepackets illustrate cases of negative and

positive displacement, respectively. For **b** $N_x = 25$, **c** $N_x = 50$, and **d** $N_x = 100$ infinite layers, collective line shifts, in units of the single-atom linewidth $\gamma$, are presented for the lattice spacing $a = 0.45\lambda$, where $\lambda$ is the resonance wavelength. The in-plane quasimomentum, indicative of the incident light's tilting angle, is varied. The collective resonance linewidth is normalised to $\gamma/(N_x - 1)$ on a colour-coded logarithmic scale. This choice highlights the linear dependence of the wavepacket lifetime, and displacement $D$, on sample thickness $a(N_x - 1)$, as alluded to in (**a**). Anomalous bright dots correspond to resonances due to array edges in the $x$-direction.

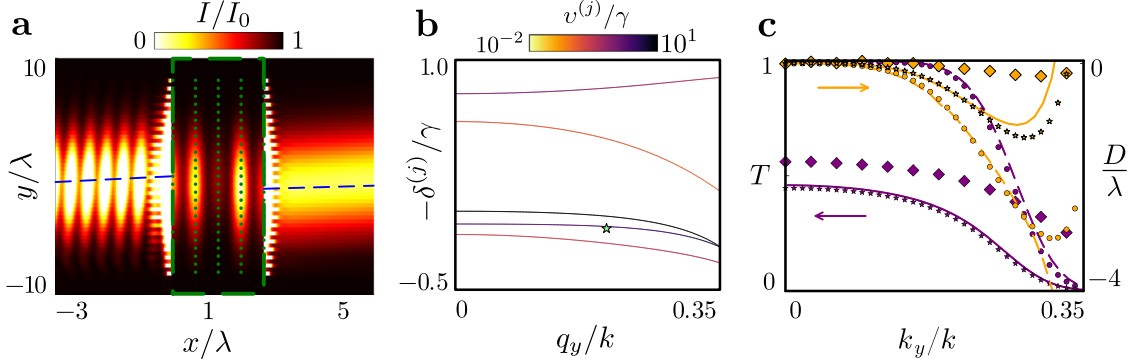

**Fig. 5 | Effects of finite lattice size and imperfections on negative refraction.**
**a** Negative refraction of light through a $5 \times 25 \times 25$ atomic array, denoted by green dots, with spacing $a = 0.68\lambda$, resonance wavelength $\lambda$, and an isolated $\sigma^+$-polarised two-level transition. The light is incident in the $xy$ plane at angle $\theta = \arcsin(0.2)$ to the lattice normal, with laser detuning $\Delta = -0.1\gamma$ from the atomic resonance, scaled by the single-atom linewidth $\gamma$. The image shows the normalised light intensity profile $I/I_0$ (outside the medium) and atomic polarisation density $|\langle \hat{\mathbf{P}}^+ \rangle|^2/|\langle \hat{\mathbf{P}}^+ \rangle|^2_{\max}$ (within the lattice delimited by the dashed green box) at plane $z = 0$, scaled by $\lambda$, where $I_0 = 2\epsilon_0 c \max_{\mathbf{r}} |\mathcal{E}^+(\mathbf{r})|^2$ represents the maximum incident intensity. $|\langle \hat{\mathbf{P}}^+ \rangle|^2/|\langle \hat{\mathbf{P}}^+ \rangle|^2_{\max}$ for point-like atoms is smoothed by convolution with a Gaussian of the root-mean-square widths $\sigma_x = 0.25a$ and $\sigma_y = 0.5a$. The blue dashed lines trace the peak light intensity. **b** Collective line shifts $\delta^{(j)}(q_y, q_z = 0)$, in units of $\gamma$, for atomic Bloch wave resonances across bands indexed by $j$ in the corresponding lattice with infinite in-

plane layers. The in-plane quasimomentum $q_y$, indicative of the incident light's tilting angle, is varied. The colour coding represents the collective resonance linewidth (see Methods), $v^{(j)}(q_y, q_z = 0)$, on a logarithmic scale, normalised to $\gamma$. In contrast with Fig. 3b, the larger lattice spacing gives rise to diffraction of phase-matched beams once $q_y \gtrsim 0.48k$, so we restrict the quasimomentum $q_y \lesssim 0.35k$ to lie well within this range. The green star denotes the dominantly excited resonance with linewidth $\simeq 0.05\gamma$ in (**a**). **c** Power transmission $T$ and lateral displacement $D$ from the centre of the layer at the exit (not the displacement of the incident beam), in units of $\lambda$, as a function of the incident light's wavevector $y$-component $k_y$. Perfect lattice with infinite layers (dashed lines) and finite-size layers (dotted lines); atomic position fluctuations with $1/e$ density width $0.074a$ about each site, obtained from stochastic simulations (diamonds); phenomenological model with corresponding imperfection parameter $\zeta_f = 0.975$ with infinite layers (solid lines) and finite-size layers (stars).

monolayer[22] that demonstrated coupling in a finite layer to collective modes resembling those of an infinite array.

Until now, we have considered steady-state responses. In Supplementary Fig. 2, we demonstrate that negative refraction can be observed even before the system reaches steady state. Supplementary Fig. 2 shows the dynamics when the light is instantaneously switched on at $t = 0$. Initially, the short time dynamics are dominated by the broad superradiant excitation, resulting in beam reflection. As time progresses, the transmission of the beam increases, accompanied by greater beam displacement and negative refraction, as the subradiant contribution is enhanced.

We next study the effect of lattice imperfections. The change in scattering resulting from imperfect filling fraction $\zeta < 1$ due to missing atoms can be quantified mean-field theoretically by taking a coarse-grained approach to each array plane[36]. For fixed atomic positions, the average atomic polarisation density then appears diminished by the factor $\zeta$. This constitutes a phenomenological model for light propagation in the presence of imperfections (Methods).

The effect of fluctuations in the atomic positions at each lattice site, which may arise from quantum or thermal fluctuations in a finite-depth optical lattice potential, can be incorporated through stochastic electrodynamics simulations. In this method, the dynamics in Eq. (1) are solved for a specific set of fixed atomic positions, stochastically sampled from the density distributions at each lattice site[49]. The expectation values are then obtained by ensemble averaging over many realisations. This approach has been formally shown to converge to the exact result in the studied cases[27,28].

For large arrays, a computationally faster, coarse-grained estimate can be obtained using the momentum-space representation of the scattering problem by incorporating smearing out of high momenta according to the atomic position uncertainty (Supplementary Note 1). This approach is adapted from the momentum-space regularisation procedure of infinite lattices[32,50]. Here, the momentum cutoff term $\exp[-q^2\eta^2/4]$, where $\eta > 0$ identifies a non-zero cutoff length, physically represents the Gaussian $\exp[-r^2/\eta^2]$ smearing of atomic positions. It can also be approximated by the phenomenological model based on the reduced polarisability, collective linewidth and line shift by the factor $\zeta_f = \exp[-k^2\eta^2/4]$ (Methods).

The few-layer finite lattice size of Fig. 5a allows us to include full stochastic simulation analysis of atomic position fluctuations. We assume Gaussian density distributions about each site with $1/e$-width $\eta = 0.074a$, which are present for moderate optical lattice depths of several hundred recoil energies, employed in light transmission experiments[22,23]. Despite the fluctuations, a coherent outgoing beam trajectory generally remains present in the realisation-averaged intensity distribution to yield a precise estimate for $D$, within the sampling error.

In Fig. 5c, we show power transmission and lateral beam displacement at varying angles of incidence in the presence of position fluctuations. For detuning $\Delta = -\zeta_f \times 0.1\gamma$, aligned with the resonance shift obtained from the phenomenological model (Methods), we find that negative refraction ($D < 0$), with significant transmission $T \gtrsim 0.3$, persists almost entirely throughout the transmission band ($\theta \lesssim 0.16 \times \pi$). Generally, reduced values of $T$ and $|D|$ result from degraded coherent coupling to phase-matched lattice modes. Apparent reductions in displacement of the total transmitted beam can also partially be attributed to contributions from incoherent scattering, which account for the increase in $T$ conversely observed towards the edge of the transmission band. The results are otherwise qualitatively in agreement with estimates of the phenomenological model in which case the predictions also remain valid for the imperfections due to missing atoms with filling fractions $\zeta = 0.975$ that are lower than achievable experimental values (e.g., above 0.99 in ref. [51]). The effect of minor imperfections is to smoothly decrease coherent transmission and the magnitude of beam displacement from their ideal values, and our findings accordingly display no sharp threshold for the emergence

of negative refraction when varying filling fraction, spatial fluctuations, or incident beam angle. Thus, even with experimentally realistic atomic position fluctuations and missing atom numbers, the investigation of negative refraction remains feasible. Whilst strong dipole-dipole interactions at very small lattice constants are expected to lead to more pronounced effects of position fluctuations, we have confirmed with exact stochastic simulations also for shorter lattice spacings $a = 0.45\lambda$, and for the $J = 0 \rightarrow J' = 1$ atomic transition, that negative refraction through five layers remains observable for $\eta/a \simeq 0.074$, with similar drops in beam transmission and displacement magnitude when targeting resonances of comparable linewidth.

As shown in Methods, the phenomenological model indicates that greater deterioration of displacement and transmission occurs for any given lattice constant when exciting subradiant eigenmodes with narrower resonances. To maintain high transmission, exciting these narrow resonances then requires tighter trapping potentials.

Previously, we linked the numerically calculated cooperative atom response to the real part $n'_{eff}$ of the effective refractive index of the sample based on the deflection of the propagating light beam. The connection between microscopic electrodynamics simulations of a granular atomic ensemble and the standard material parameters of macroscopic continuous media electrodynamics can be further explored by considering qualitative descriptions for the imaginary component $n''_{eff}$ of the effective refractive index ($n_{eff} = n'_{eff} + in''_{eff}$). We can estimate the attenuation length of a beam due to imperfections using a mean-field approach in the phenomenological model that depends on the out-of-plane group velocity, as demonstrated in Supplementary Fig. 3. This relationship suggests resonant eigenmodes with large out-of-plane group velocities may be preferentially targeted to increase visibility of the transmitted beam.

Our calculations are based on the low light intensity limit, where each atom responds linearly to light. However, in Supplementary Fig. 4, we demonstrate that negative refraction is not exclusively a phenomenon of linearly responding systems and can persist beyond the low light intensity limit. We incorporate beyond low light intensity effects within a mean-field approximation, which neglects quantum correlations between different atoms[28]. This approach has provided good qualitative descriptions of optical responses at all intensities outside the regimes of phase transitions[52]. We find that, similar to the effects of imperfect filling fractions and position fluctuations, excitations in the studied example are reduced due to nonlinearities, resulting in decreased coherent light transmission and beam displacement.

## Discussion

We have conducted large atomic-scale simulations to demonstrate the negative refraction of light in atomic media. These simulations are based on a methodology that has been shown to accurately describe the experiments on atoms illuminated with resonant light in optical lattices[22,23]. The essentially exact treatment of light-atom coupling establishes a direct link between the microscopic quantum properties of atoms and material bulk parameters of macroscopic electromagnetism.

High-transmission negative refraction is observed in a lattice over a broad range of incident angles ($\pi/6 \lesssim \theta \lesssim \pi/2$ in Fig. 1) across the transmission band, even without fine-tuning the lattice geometry or optimising parameters. These conclusions hold equally for beams incident in both the $xy$ and $xz$ planes, demonstrating that negative refraction is present in a 3D angular range. We find that negative refraction can generally be achieved in few-layer and many-layer subwavelength lattices, regardless of the lattice constant, both with equal or unequal subwavelength spacing, and even when the lattice constant in one dimension exceeds the resonance wavelength.

In the studied examples, negative refraction remain robust in the presence of lattice imperfections due to position fluctuations in optical lattices with moderate depths (several hundred recoil energies, as in light transmission experiments[22,23]), and due to missing atoms with

filling fractions $\zeta = 0.975$ in Fig. 5 that are lower than achievable experimental values. We have derived a general condition, $\zeta v \gtrsim (1-\zeta)\gamma$, in terms of collective and single-atom linewidths, to observe the relevant resonances (Methods).

Atoms can now routinely be prepared in Mott-insulator states in optical lattices. Recent experiments include observations of anti-ferromagnetism in a 3D lattice of 800,000 sites[21]. The atomic arrays offer several distinct advantages for the studies of light propagation. Unlike cold, random atom clouds, they allow for well-defined medium boundaries. Technology for trapping atoms in periodic arrays is advancing rapidly[22,23,53–59], offering a broad range of potential applications[20]. Cold atoms in optical lattices serve as a promising platform for quantum simulators[60]. In marked contrast to solid-state metamaterials[61], quantum interfaces between atoms and light are already well established, with the potential for the arrays to operate in the quantum regime[52,62] at the single-photon level and serve as quantum networks[63]. The atoms also exhibit long coherence times and are free from non-radiative absorption losses. Additionally, a well-developed atomic physics technology exists to control and manipulate atomic sites and internal atomic levels that possess precise resonance frequencies, free from manufacturing imperfections. Atomic arrays with sub-wavelength spacings experience strong multiple scattering, enabling non-local collective and nonlinear responses, while strong collective effects in metamaterials typically require special circumstances[64,65]. This makes atomic arrays an exciting platform for exploring, e.g., topologically non-trivial phases[66] and time-varying optical phenomena[67,68] with negative refraction. A microscopic description of atomic arrays as an ensemble of stationary atoms interacting with light at the low intensity limit—and thus our formalism—is exact[27,28] for the studied atomic level schemes, eliminating the need to approximate the resonators' internal structure, as is necessary in collectively responding metamaterials[65,69]. This enables both accurate and computationally efficient design of optical devices and functionalities in increasingly large-scale atomic systems realised in the state-of-the-art experiments.

## Methods
### Calculation of the scattered field
Here, we expand upon the formalism enabling the calculation of induced dipoles and the scattered field for arrays of infinite extent. A detailed discussion of light-atom interactions may be found in the Supplementary Note 1. We substitute the Bloch wave representation $\mathcal{P}^{(\ell,j)} = \sum_{\mathbf{q}_\parallel} \mathcal{P}_\ell(\mathbf{q}_\parallel) e^{i\mathbf{q}_\parallel \cdot \mathbf{r}_{\parallel j}}$ into the multiple scattering relation Eq. (1), to obtain a momentum-space counterpart describing scattering of in-plane excitations between layers,

$$\dot{\mathcal{P}}_\ell(\mathbf{q}_\parallel) = (i\Delta - \gamma)\mathcal{P}_\ell(\mathbf{q}_\parallel) + i\mathcal{R}_\ell^+(\mathbf{q}_\parallel) + i\xi \sum_{m=0}^{N_x-1} \mathrm{G}^{\mathrm{L}}(x_\ell - x_m, \mathbf{0}; \mathbf{q}_\parallel)\mathcal{P}_m(\mathbf{q}_\parallel), \quad (2)$$

where $\mathcal{R}_\ell^+(\mathbf{q}_\parallel) = \mathcal{D}\mathcal{E}_\ell^+(\mathbf{q}_\parallel)/\hbar$ are the in-plane Fourier components of the incident field Rabi frequency when restricted to the plane $x = x_\ell$. The layer propagator $\mathrm{G}^{\mathrm{L}}(x, \mathbf{r}_\parallel; \mathbf{q}_\parallel)$ plays the role of a dipole radiation kernel for an infinite layer located at $x=0$ and excited with in-plane quasimomentum $\mathbf{q}_\parallel$. For $x=0$, $\mathbf{r}_\parallel = \mathbf{0}$ ($\ell = m$ in Eq. (2)), $\mathrm{G}^{\mathrm{L}}$ describes the light-mediated interaction between atoms within each 2D layer[70,71] and is included even in the case of a single layer. The layer propagator is defined in each case by a lattice sum,

$$\mathrm{G}^{\mathrm{L}}(x, \mathbf{r}_\parallel; \mathbf{q}_\parallel) = \sum_j e^{i\mathbf{q}_\parallel \cdot \mathbf{r}_{\parallel j}} \mathrm{G}(\mathbf{r}_\parallel - \mathbf{r}_{\parallel j} + x\hat{\mathbf{e}}_x), \quad x \neq 0, \quad (3)$$

$$\mathrm{G}^{\mathrm{L}}(0, \mathbf{0}; \mathbf{q}_\parallel) = \sum_{j \neq 0} e^{i\mathbf{q}_\parallel \cdot \mathbf{r}_{\parallel j}} \mathrm{G}(\mathbf{r}_{\parallel j}), \quad (4)$$

for the dipole radiation kernel[31],

$$\mathrm{G}_{\nu\mu}(\mathbf{r}) = \left[ \frac{\partial}{\partial r_\nu} \frac{\partial}{\partial r_\mu} - \delta_{\nu\mu} \boldsymbol{\nabla}^2 \right] \frac{e^{ikr}}{4\pi r} - \delta_{\nu\mu}\delta(\mathbf{r}). \quad (5)$$

The momentum summation within each layer, encapsulated in Eqs. (3) and (4), provides numerically efficient computation: solving for the steady-state polarisation amplitudes $\mathcal{P}_\ell(\mathbf{q}_\parallel)$ of $N_x$ atomic layers amounts to inverting, for each $\mathbf{q}_\parallel$, a $3N_x \times 3N_x$ block matrix. $\mathrm{G}^{\mathrm{L}}$ is used in Eq. (2) to solve for the induced dipoles under light illumination and to obtain the scattered field by decomposing into radiation contributions for each layer $\ell$ and polarisation Bloch wave $\mathcal{DP}_\ell(\mathbf{q}_\parallel)$,

$$\epsilon_0 \langle \hat{\mathbf{E}}_s^+(\mathbf{r}) \rangle = \sum_{\mathbf{q}_\parallel} \sum_{\ell=0}^{N_x-1} \mathrm{G}^{\mathrm{L}}(x - x_\ell, \mathbf{r}_\parallel; \mathbf{q}_\parallel) \mathcal{DP}_\ell(\mathbf{q}_\parallel). \quad (6)$$

These are used to generate the intensity profiles in Figs. 1b, 2, and 5a. Power transmission $T$, presented in Figs. 1d, e and 5c, is calculated using the total field intensity, with the expectation values taken over fluctuating atomic positions,

$$T = \frac{\int \sum_\mu \langle \hat{\mathbf{e}}_\mu \cdot \hat{\mathbf{E}}^+(\mathbf{r}) \hat{\mathbf{E}}^-(\mathbf{r}) \cdot \hat{\mathbf{e}}_\mu^* \rangle d\Omega}{\int \boldsymbol{\mathcal{E}}^+(\mathbf{r}) \cdot \boldsymbol{\mathcal{E}}^-(\mathbf{r}) d\Omega}, \quad (7)$$

for solid angle elements $d\Omega$ enclosing the incident and transmitted beams, where the summation is over the three orthogonal polarisations. To analyse focussed transmission, we assume a small collection surface in the plane $x = a(N_x - 1) + 2\lambda$, extending across $-10\lambda \leq y, z \leq 10\lambda$.

### Calculation of the excitation band structure
The multiple scattering relation between atomic layers, Eq. (2), can be cast in the form

$$\dot{\mathbf{b}}(\mathbf{q}_\parallel) = i[\mathcal{H}(\mathbf{q}_\parallel) + \delta\mathcal{H}]\mathbf{b}(\mathbf{q}_\parallel) + \mathbf{f}(\mathbf{q}_\parallel), \quad (8)$$

with $\mathbf{b}_{3m-1+\nu}(\mathbf{q}_\parallel) = \mathcal{P}_{m\nu}(\mathbf{q}_\parallel)$, $\mathbf{f}_{3m-1+\nu} = i\hat{\mathbf{e}}_\nu^* \cdot \mathcal{R}_m^+(\mathbf{q}_\parallel)$, whilst the diagonal matrix $\delta\mathcal{H}$ contains $\Delta$. The collective excitation eigenmodes and the resulting band structure in the case of isotropic polarisation is determined by the eigenvectors of the $3N_x \times 3N_x$ matrix $\mathcal{H}$

$$\mathcal{H}_{3n-1+\nu, 3m-1+\mu}(\mathbf{q}_\parallel) = \xi \mathrm{G}_{\nu\mu}^{\mathrm{L}}(x_n - x_m, \mathbf{0}; \mathbf{q}_\parallel) + i\gamma \delta_{\nu\mu}\delta_{nm}. \quad (9)$$

The corresponding eigenvalues $\delta^{(j)}(\mathbf{q}_\parallel) + iv^{(j)}(\mathbf{q}_\parallel)$ comprise the collective line shifts $\delta^{(j)}(\mathbf{q}_\parallel)$ and linewidths $v^{(j)}(\mathbf{q}_\parallel)$ for each in-plane quasimomentum $\mathbf{q}_\parallel$ in band $j$. The $N_x$ layer degrees of freedom and three orthogonal dipole orientations available to the $J = 0 \to J' = 1$ transition together yield $3N_x$ eigenvalues for each $\mathbf{q}_\parallel$, which we calculate to give the $3N_x$ bands presented in Fig. 3a, b. To calculate the band structure in the case of two-level atoms (Fig. 5b) with the unit dipole vector $\hat{\mathbf{e}}_+$, we have the matrix $\mathcal{H}_{n,m}$, with $N_x$ eigenvalues, for each $\mathbf{q}_\parallel$.

### Modelling imperfections through diminished atomic polarisation density
We use diminished atomic polarisation density by a factor of $\zeta$ to phenomenologically represent fractional filling $\zeta < 1$. This mean-field averaging approach can describe the impact of missing atoms on light transmission through atomic arrays for $\zeta \simeq 1$, as numerically demonstrated in ref. 36. In the calculation of the collective response in Eqs. (2) or (8), this approach amounts to using $\mathcal{H}^*$ in place of $\mathcal{H}$ for the band structure,

$$\mathcal{H}_{3n-1+\nu, 3m-1+\mu}^*(\mathbf{q}_\parallel) = \zeta \xi \mathrm{G}_{\nu\mu}^{\mathrm{L}}(x_n - x_m, \mathbf{0}; \mathbf{q}_\parallel) + i\gamma \delta_{\nu\mu}\delta_{nm}. \quad (10)$$

Examining the eigenvalues of $\mathcal{H}^*$ reveals the collective resonance line shifts $\zeta\delta^{(j)}$ and linewidths $\zeta\upsilon^{(j)} + (1-\zeta)\gamma$ moving towards the single-atom values, which we illustrate in Supplementary Fig. 5, due to the reduced effect of light-mediated interactions through $\zeta G^L$. This can be straightforwardly demonstrated using the single-layer form of Eq. (2)

$$\dot{\mathcal{P}}_0(\mathbf{q}_\parallel) = (i\Delta - \gamma)\mathcal{P}_0(\mathbf{q}_\parallel) + i\mathcal{R}_0^+(\mathbf{q}_\parallel) + i\zeta\xi G^L(\mathbf{0},\mathbf{0};\mathbf{q}_\parallel)\mathcal{P}_0(\mathbf{q}_\parallel)$$
$$= [i\Delta + i\zeta\delta_s(\mathbf{q}_\parallel) - \gamma - \zeta\tilde{\gamma}_s(\mathbf{q}_\parallel)]\mathcal{P}_0(\mathbf{q}_\parallel) + i\mathcal{R}_0^+(\mathbf{q}_\parallel), \quad (11)$$

leading to the Lorentzian steady-state solution of the single-layer excitation amplitude:

$$\mathcal{P}_0(\mathbf{q}_\parallel) = \frac{-\mathcal{R}_0^+(\mathbf{q}_\parallel)}{\Delta + \zeta\delta_s(\mathbf{q}_\parallel) + i[\zeta\upsilon_s(\mathbf{q}_\parallel) + (1-\zeta)\gamma]}, \quad (12)$$

expressed in terms of the collective linewidth $\upsilon_s(\mathbf{q}_\parallel) = \gamma + \tilde{\gamma}_s(\mathbf{q}_\parallel)$ and line shift $\delta_s(\mathbf{q}_\parallel)$, respectively, for the single layer. Analogous to the resulting transmission amplitude of a single atomic layer,

$$t = \frac{\Delta + \zeta\delta_s + (1-\zeta)\gamma}{\Delta + \zeta\delta_s + i(\gamma + \zeta\tilde{\gamma}_s)}, \quad (13)$$

we find that the transmission in the multilayer scenario similarly diminishes by the factor $(1-\zeta)\gamma$. This observation aligns with our analysis; generally, transmission notably falls from the ideal $\zeta = 1$ when $\zeta\upsilon^{(j)}(\mathbf{q}_\parallel) \lesssim (1-\zeta)\gamma$.

The averaged effect of the atomic position fluctuations on the excitation band structure can likewise be estimated by the similar diminished light-mediated interactions[32,50]. This method has previously also been applied to model the optical responses of atom arrays in the presence of position uncertainty[72]. In such instances, Gaussian $\exp(-r^2/\eta^2)$ position fluctuations at each lattice site in a finite-depth trap are represented by a momentum-space band-structure summation with a finite cutoff length $\exp(-k^2\eta^2/4)$, mirroring the Fourier transform of the lattice site density distribution (Supplementary Note 1). It is analytically demonstrated that this effectively scales the light-mediated interactions by a factor of $\zeta = \exp(-k^2\eta^2/4)$[32,50]. We have both numerically and analytically corroborated that also in our system of a finite number of layers introducing the phenomenological model of diminished propagator $\zeta G^L$ closely agrees with the results derived using the momentum-space band-structure summation with a finite cutoff length. Whilst either of these coarse-grained estimates offer a computationally efficient alternative to exact stochastic electrodynamic simulations for calculating light propagation and band structure in larger atomic lattices, the significant discrepancies in transmission even for weak imperfections $\zeta = 0.975$ (Fig. 5c, Main Text) highlight their limitations in capturing the full complexity of light propagation in disordered media, where they generally only provide qualitative estimates.

## Data availability
Data is available at https://doi.org/10.5281/zenodo.14271057.

## Code availability
The code is available upon request.

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

## Acknowledgements

We are grateful for fruitful discussions with T. Busch, C. Gross, W. J. Munro, C. Simovski and S. Tretyakov. We acknowledge financial support, in part, from Moonshot R&D, JST JPMJMS2061 (L.R.), the UK EPSRC, Grants No. EP/S002952/1 (J.R.) and No. EP/W005638/1 (K.E.B.).

## Author contributions

L.R. and J.R. formulated the theory and methodology. L.R. produced the figures and performed the numerical simulations, with supporting investigations by J.R.; K.E.B. confirmed numerical finite-size results with independent codes. J.R. instigated the project. All authors discussed the results and commented on the manuscript.

## Competing interests

The authors declare no competing interests.
