## [Transparent Peer Review file · Nature Communications]

Negative refraction of light in an atomic medium

Corresponding Author: Professor Janne Ruostekoski

Version 0:

Reviewer comments:

Reviewer #1

(Remarks to the Author)

In their manuscript "Negative refraction of light in an atomic medium" L. Ruks, K. E. Ballantine, and J. Ruostekoski perform a detailed analysis of the light propagation through a three dimensional arrangement of atom arrays exhibiting cooperative light-induced dipole-dipole interactions. They find that such an arrangement can result in negative refraction of light, i.e. light is refracted opposite than expected from standard linear optics. The authors perform large scale simulations based on a momentum space representation of the collective modes in the low intensity approximation for each atomic layer, which they assume to be infinite in the y-z plane. This enables them to construct an effective light propagator for each layer which allows a detailed and accurate study of the phenomenon. In the last section the authors also analyze the role of disorder and finite array sizes in the y-z plane.

The performed analysis and methodology seems correct and the results are interesting. Nevertheless, I cannot recommend the manuscript for publication in Nature Communication since I don't think it meets the criteria of significance and novelty required for this particular journal as I will elaborate more below.

1. I think the manuscript is a very nice addition to a body of work on cooperative light scattering in subwavelength arrays (see e.g. Refs. [45]-[48] in the manuscript) that was established over the past decade. While a detailed study of light propagation through layers of arrays is warranted, I don't see the degree of innovation in this study. It is known that changing the atomic geometry allows band structure engineering so I think the claim that this manuscript establishes "a link between the microscopic quantum properties of atoms and material bulk parameters of macroscopic electromagnetism" does not hold.

2. The authors argue that the presented setup would have benefits over metamaterial engineering approaches since negative refraction only occurs due to the collective response based on the atomic lattice geometry. However, a realization of the studied system is highly non-trivial and is beyond any current technical capabilities in state-of-the-art AMO experiments. In fact, only very few experiments currently reached the limit of realizing subwavelength arrays in 2D, let alone stacking them in 3D. Hence, the authors should definitely discuss the feasibility of generating such a geometry given current technology in more detail.

3. Another criticism I have on the paper is that the entire analysis is performed in the low driving limit, which makes the momentum space representation the authors use exact. This approach has been extensively exploited in the literature over the past years, and is known as an efficient way to characterize light propagation through cooperative arrays. However, it would be valuable to include a discussion or preliminary analysis of how higher light intensities and resulting nonlinearities might affect the negative refraction, potentially altering the conclusions drawn under the low-intensity approximation.

In summary, while the manuscript presents a thorough analysis of negative refraction in atomic media, the novelty and experimental feasibility do not meet the high standards required for publication in Nature Communications. Therefore, I recommend that the authors consider submitting this work to a more specialized journal.

Reviewer #2

(Remarks to the Author)

The manuscript "Negative refraction of light in atomic medium" presents a rigorous numerical analysis of the idea to use an ordered atomic array with sub-wavelength spacing to generate a negative index of refraction of the atomic medium at optical

frequencies. The negative refraction angle results from cooperative effects between the atoms for some light-atom detunings and certain ranges of incidence angle. The authors cleverly reduce the complexity of the problem by decomposing the atomic polarization dynamics within a single atomic layer into Bloch waves propagating along in-plane wavevectors. This allows them to simplify the spectral analysis of the coupled-oscillator problem, which leads to a clear picture of dispersion band folding, resulting in the effective negative refractive index. The numerical analysis is further extended to include typical experimental imperfections such as finite temperature of the atoms, disorder in the trap array positions, and missing atoms. The authors conclude that even in the presence of these imperfections, the negative refraction angle should be observable with state-of-the-art experimental techniques.

The article is well written and logically structured, and easy to follow. Cited references establish a good context for the presented work, and support the arguments made by the authors. While we cannot directly verify the calculations in the manuscript, the approach taken by the authors is reasonable, and the results appear valid. The topic of a negative index of refraction at optical frequencies is an interesting one, and the proposed approach may be superior to nanofabricated systems. We believe that the manuscript will be of interest to the readership of Nature Communications, and we recommend publication. However, we ask the authors to comment on the following questions:

1. The authors show the negative displacement of an incoming beam for a certain range of input angles. However, the discussion of the negative index of refraction in the introduction is quite general, and at least for the simplest applications one would want the negative index to hold for all (or at least, most) incidence angles. To what extent do the demonstrated results actually generalize to a situation with a negative index of refraction, rather than just some unusual dispersion? Is the system as calculated useful for applications? The authors criticize in the introduction that systems with photonic crystals “are limited to narrow range of incidence angles and frequencies” – how does the proposed system compare to photonic crystals?
2. The article offers a well-reasoned numerical study, yet it lacks a tangible and intuitive microscopic explanation for the observed effects. Can the authors think of an intuitive mechanism behind the phenomenon studied in the article? (A microscopic picture could perhaps be elucidated based on the eigenvectors in the spectral analysis.) Could the authors attempt to explain the effects they observe based on physics arguments, perhaps with some pictures of microscopic polarization profiles? Is such explanation feasible in light of cooperative effects, and if not, why not? An answer to these questions is not a requirement for publication in Nature Communications, but such an explanation would make the manuscript much stronger, and more interesting to a general audience.
3. Could the authors provide a paragraph briefly summarizing the parameter regimes and imperfection thresholds for observing the negative refraction? How does the refraction deteriorate when approaching the thresholds (e.g. in the required atom filling)? Are there some intuitive guidelines for picking the best experimental parameters to observe the phenomenon? Such a summary would be very helpful to experimentalists considering a realization of the proposal.
4. The paragraphs explaining the spectral analysis and the methodology of relating its results to the beam displacements (lines 181-207) could appear earlier in the article, e.g. after line 115. This would make it easier to follow the article.
5. The colormap in Fig. 1c could be chosen better to highlight the transition from positive to negative displacements. A simple blue-white-red map would work well.

Reviewer #3

(Remarks to the Author)

Reviewer #4

(Remarks to the Author)

In the manuscript, the authors theoretically investigated the propagation of light through a 3D atomic array and showed that, under certain conditions, this peculiar atomic medium possesses a negative refractive index. Initially, the authors considered the ideal case of an infinite and defect-free system, but they also moved to the case of finite size, where boundary effects can have important consequences. In both cases, conditions for negative refractive index are found. Interestingly, these results seem to be robust to defects, which means that the phenomenon under investigation might be observable in the state-of-the-art experiments.

I find the manuscript clear, the topic interesting, and the results new and fascinating. Furthermore, the theoretical analysis is deep and precisely carried out. For all these reasons, I recommend publication in Nature Communications. However, I would like to ask the authors to consider the following questions:

1- Even if subwavelength spaced atomic arrays have been realized (see reference 20 and 21), their experimental realization remains challenging. At line 177, the authors claim that it is not important to have equal in-plane and out-of-plane lattice constant for having negative refraction. I wonder if one of these conditions can be relaxed, for example, by having a distance between layers greater than λ (or the opposite configuration). Can similar effects be observed when working with layers of randomly distributed 2D atom clouds?

2- Although the general interest of the topic treated by the manuscript, it is not clear to me what the prospects are. In the introduction, the authors claim that metamaterials can be useful for the realization of superlensing, for example. Is this a general statement? Or can the specific system investigated be used for this purpose (or any other technological application)?

3- In the case of a finite system, the authors report their results in Fig 2. According to Fig2a, in steady state, the light intensity is maximized around two layers. This behavior is qualitatively different from that reported in Fig1 for the infinite system, where instead all layers are equally "illuminated". Is there any intuitive explanation for this? Is it related to the number of layers or to the finite size of the sample?

4- Related to the last point, the authors only considered 5 layers for finite size systems. It is not clear to me how the number of layers matters. According to Fig. 3, apart from the initial 2-3 layers, the displacement D seems to increase linearly with the number of layers, is this correct? I find it difficult to understand Fig 3. Firstly, is it evaluated for a finite or infinite size array? Could it be simpler to plot the displacement D as a function of the number of layers, for example, for two or three values of k_y ?

5- The simulations for the finite size system were carried out considering a probe beam with a width of 5λ . What exactly do you mean by that? Is it the waist of the beam or its diameter? I'm also wondering about the effect of beam size for a finite system. Indeed, I would expect that a too much focused probe would lead to a reduction of the collective effects, since the light effectively experiences a smaller number of atoms. On the other hand, too large a beam waist would lead to an important effect of the edges of the atomic distribution, again weakening the collective behavior. Is my intuitive picture correct? If so, is there an optimal beam size to enhance the negativity of the refractive index of the material? I think this point could be important in view of a future experimental realization.

6- All the results presented here are calculated in the low light limit, where each of the emitters behaves as a classical dipole. What is expected beyond this limit? How much the negativity of the refractive index is robust to an increasing driving light? Can the quantum nature of a saturated dipole introduce some interesting properties into the system?

7- Finally, I have a question out of curiosity. Most of the effects considered here have been evaluated in the steady state. However, I was wondering if the negativity of the refractive index could also manifest itself in the time domain, for example by studying the modification of the temporal profile of a short light pulse by the propagation of the medium. Could this be another possible observable to measure the negativity of the refractive index? More generally, are there any related collective effects affecting the dynamical evolution before the establishment of the steady state?

8- Even if the number of papers related to this topic is large, I would ask the authors to consider these papers that I find related to the topic of the manuscript:

- Bekenstein, R. et al. "Quantum metasurfaces with atom arrays." *Nat. Phys.* 16, 676–681 (2020).

- Bettles, R., et al., "Enhanced optical cross section via collective coupling of atomic dipoles in a 2D array." *Physical review letters* 116.10 (2016): 103602.

- Solomons, Y., et al., . "Universal approach for quantum interfaces with atomic arrays." *PRX Quantum* 5.2 (2024): 020329.

- Asenjo-Garcia A., et al, Exponential Improvement in Photon Storage Fidelities Using Subradiance and "Selective Radiance" in Atomic Arrays, *Physical Review X* 7, 031024

9- A final detail: I'm not sure if it is an issue of the version I have, but the quality of the figures is very poor

Version 1:

Reviewer comments:

Reviewer #1

(Remarks to the Author)

I would like to thank the authors for their detailed response to my comments and for the additional data they provided to clarify point 3 in my original response. As already pointed out in my original report I consider the results presented in this paper of overall high quality and the additional data also removed some of my concerns regarding the restriction to the single excitation subspace. I appreciate the authors' arguments about the significance of this work, which indeed align with the assessment by referees 2, 3, and 4. Hence, my assessment of the potential impact of this work was based on limited knowledge about the decade long struggle in designing a medium with a negative refractive index for light. In light of these new insights I recommend this manuscript for publication in *Nature Communication*.

I still slightly disagree with the authors' response regarding the experimental realizability of the proposed setup. While the works by Immanuel Bloch, Jan-Wei Pan, Jun-Ye, etc. they mention indeed show that sub wavelength lattices are possible in modern cold atom experiments (even in 3D). The level of band structure engineering required to realize the effects presented in this work, will still need a high degree of control on the positional configuration of the individual atoms, which I still see as very challenging even within these setups. This is not a point, that will change my overall assessment, but it is worth pointing out that more experimental progress has to be made until this proposal can be fully realized in a lab.

Reviewer #2

(Remarks to the Author)

We thank the authors for the careful consideration of our comments, as well as those of the other referees. We find that the

presentation of the authors' work has greatly improved owing to the revised figures and other changes. The choice of the colormap for Fig. 1c is suitable and it highlights that the range of incident angles for which the negative refraction can be observed is indeed very broad. The cartoon in Fig. 2 that illustrates a propagating atomic polarization wavepacket helps to make the concepts appearing in the calculations more tangible. We understand the journal's policy on the article's structure and therefore find the decision to keep the order of text unchanged well justified. The updates to the 'Discussion' section are also satisfactory.

We are happy with the manuscript in its current form and recommend its publication in Nature Communications. This is important theoretical work that is likely to trigger experimental efforts to observe the negative index of refraction.

Reviewer #3

(Remarks to the Author)

Reviewer #4

(Remarks to the Author)

Firstly, I would like to thank the authors for the effort they have made to answer the questions raised and to modify the text accordingly. I find the new version of the manuscript clearer, and I believe that the addition of Figure 4 further strengthens the results obtained. For these reasons, I recommend that the manuscript be published.

Response to Reviewer 1

In their manuscript "Negative refraction of light in an atomic medium" L. Ruks, K. E. Ballantine, and J. Ruostekoski perform a detailed analysis of the light propagation through a three dimensional arrangement of atom arrays exhibiting cooperative light-induced dipole-dipole interactions. They find that such an arrangement can result in negative refraction of light, i.e. light is refracted opposite than expected from standard linear optics. The authors perform large scale simulations based on a momentum space representation of the collective modes in the low intensity approximation for each atomic layer, which they assume to be infinite in the y-z plane. This enables them to construct an effective light propagator for each layer which allows a detailed and accurate study of the phenomenon. In the last section the authors also analyze the role of disorder and finite array sizes in the y-z plane. The performed analysis and methodology seems correct and the results are interesting. Nevertheless, I cannot recommend the manuscript for publication in Nature Communication since I don't think it meets the criteria of significance and novelty required for this particular journal as I will elaborate more below.

1. I think the manuscript is a very nice addition to a body of work on cooperative light scattering in subwavelength arrays (see e.g. Refs. [45]-[48] in the manuscript) that was established over the past decade. While a detailed study of light propagation through layers of arrays is warranted, I don't see the degree of innovation in this study. It is known that changing the atomic geometry allows band structure engineering so I think the claim that this manuscript establishes "a link between the microscopic quantum properties of atoms and material bulk parameters of macroscopic electromagnetism" does not hold.

We thank Reviewer 1 for their commitment to refereeing our manuscript. We first want to emphasise that the essential result of the study is the demonstration of negative refraction of light in atomic media under experimentally realisable conditions, without the need for metamaterials (the experimental feasibility of the lattice configuration is discussed in the following response). This has never been achieved before due to significant limitations of optical properties of natural materials. These limitations led to the development of artificially manufactured metamaterials. The main driving force behind this development has been negative refraction of light, with enticing applications such as superlensing, allowing focussing and imaging beyond the diffraction limit, as well as transformation optics.

Despite the significant interest in negative refraction, intrinsic non-radiative damping and ever-present fabrication imperfections of resonators in metamaterials continue to cause considerable

losses at optical frequencies, severely limiting their applications. This remains true even 24 years after Pendry’s seminal proposal on diffraction-free superlensing (Ref. [9], with nearly 16,000 citations on Google Scholar). In contrast, natural media made of atoms do not suffer from these imperfections, where the pristine conditions could be crucial for investigating and developing applications enabled by negative refraction.

Our analysis is based on microscopic atom-by-atom simulations that are exact for stationary atoms in the low light intensity limit – a condition that has been shown to describe very accurately the experiments on atoms illuminated with resonant light in optical lattices by Immanuel Bloch’s group Refs. [22,23], verifying their predictive power (see also, e.g., the recent light scattering experiments on randomly distributed atoms in arXiv:2409.04148, which is now a new reference [29]). With these simulations, we demonstrate high-transmission negative refraction that is resilient to lattice imperfections, such as missing atoms and position uncertainty, over a range of incident angles, different atomic level structures and lattice constants, also with the lattice parameters of Ref. [22].

Our work thus has the potential to break a decades-long impasse in practically realising negative refraction of light by using an atomic medium, without metamaterials, and to prompt immediate experimental activity towards its first observation.

We note that this significance and novelty in our work was acknowledged by the other Reviewers 2, 3, and 4.

In the 3D atomic medium, the development of an effective light propagator, the analysis of disorder, the description of band structure engineering, and other methodologies – which Reviewer 1 acknowledges – are secondary, although in our opinion also significant, achievements. Theoretical studies of atomic arrays interacting with light have almost solely focused on 2D arrays and our analysis specific to 3D systems is non-trivial.

Regarding the link between the microscopic quantum properties of atoms and material bulk parameters of macroscopic electromagnetism, it is important to highlight that there has been intense ongoing debate and controversy over how even the most basic concepts of macroscopic electromagnetism, such as refractive index and electric susceptibility, emerge from the microscopic principles of cooperatively interacting atoms. Several studies (e.g., Refs. [42-45]) have shown that in cold and dense atomic ensembles (even classically) light-induced strong interactions between the atoms result in a significant failure of the century-old standard electrodynamics of continuous media.

Standard macroscopic electrodynamics, along with its bulk medium parameters, is based on a

continuous effective-medium mean-field theory, which assumes that each atom interacts with the average behaviour of its surrounding atoms. This approach erases spatial information about individual atoms – and the corresponding details of the position-dependent resonant dipole-dipole interactions – resulting in approximations in the calculations of the optical response. Notable consequences of the breakdown of continuous media approaches include the absence of density-dependent resonance shifts (the Lorentz-Lorenz shift etc.; Refs. [42-44], Dalibard’s group experiments) and the emergence of a maximum refractive index, Ref. [45]. In thermal gases and in solids, continuous media electrodynamics is restored due to suppression of cooperative scattering.

While many of these questions remain unresolved, we extract a refractive index from the microscopic simulations, accompanied by a simple phenomenological explanation that links atomic quantum physics to this fundamental material parameter of macroscopic electromagnetism. We disagree with an interpretation of this process as straightforward, as it emerges from the *transverse* band structure we calculate using the specific momentum-space representation of the infinite-planar-layer analysis. Furthermore, we believe that the novel connection we have drawn from Dicke subradiance to enhanced refraction is significant.

To smoothly facilitate the related discussion in the Main Text, we have moved Supplementary Fig. 2 of the original submission, illustrating the emergence of refractive index from the microscopic principles in atomic arrays, to a new Fig. 4 (b,c,d). In this figure, the number of layers is varied from 25 to 100, effectively maintaining the macroscopic refractive response while the collective transverse dispersion band density increases. This result is possible because the collective linewidth scaling, $\nu \propto 1/N_x$, of resonant and phase-matched excitations precisely compensates for the increasing layer number N_x , such that $(N_x - 1)\nu$ remains finite for large N_x . This derivation of the scaling behaviour of the optical response as a function of the sample thickness therefore achieves the macroscopic ‘bulk’ behaviour limit of light refraction and directly demonstrates a linear dependence of the transverse displacement on thickness in macroscopic media. We have further expanded this discussion in the modified version of the manuscript.

We would again, however, like to stress that our main result is the broad demonstration of negative refraction in atomic arrays, which provide several advantages over artificially manufactured metamaterials, as discussed extensively in the “Discussion” section of the manuscript. As also recognized by Reviewer 1, atoms offer a well-defined microscopic description, eliminating the need to approximate the resonators’ internal structure, as is necessary in collectively responding metamaterials. Advanced atomic physics technologies allow for precise control and manipulation of atoms

with light, quantum interfaces between light and atoms are well established, and the atoms can exhibit strong nonlinearities, making them highly versatile. Moreover, atomic systems are inherently clean, with long coherence times and free from manufacturing imperfections and non-radiative absorption losses that are typically present in metamaterials.

2. The authors argue that the presented setup would have benefits over metamaterial engineering approaches since negative refraction only occurs due to the collective response based on the atomic lattice geometry. However, a realization of the studied system is highly non-trivial and is beyond any current technical capabilities in state-of-the-art AMO experiments. In fact, only very few experiments currently reached the limit of realizing subwavelength arrays in 2D, let alone stacking them in 3D. Hence, the authors should definitely discuss the feasibility of generating such a geometry given current technology in more detail.

We respectfully disagree that realising subwavelength arrays of atoms is beyond the current technical capabilities in AMO experiments. The experimental creation of atomic Mott-insulator states in optical lattice potentials is a 22-years-old technology and nowadays already well-established standard technique in laboratories worldwide. A 3D Mott-insulator was observed even in fermionic atoms already 16 years ago [Nature **455**, 204 (2008)]. The experimental focus has shifted to more advanced challenges (in which cases the Mott-insulator states are implicitly realised), such as the local control of atoms, the preparation of strongly correlated states that are considerably more difficult to achieve (e.g., antiferromagnetism in fermionic atoms, as discussed below), and single-site occupation control and resolution imaging.

Subwavelength Mott-insulator states in lattices can be prepared in both 3D or 2D configurations. A 2D lattice can be obtained, for instance, by combining an optical lattice potential in a plane with a tight perpendicular light sheet potential, or by emptying all but one layer of a 3D lattice. The motivation for many of the experiments to focus on 2D lattices in particular is to implement a single-site resolution imaging of atoms.

An example of the experimental progress can be seen in the recent work by the Jian-Wei Pan group, which recently reported the observation of the antiferromagnetic (Néel) phase transition in a 3D subwavelength uniform optical lattice of fermionic atoms with approximately 800,000 sites [Nature **632**, 267 (2024), now Ref. [21]]. This phase transition occurs at significantly lower temperatures and under more challenging conditions than the Mott-insulator transition, when the Mott-insulator state undergoes further spin ordering due to the dominating influence of quantum magnetism. Theoretical proposals also exist for producing subwavelength optical tweezer arrays

[see, e.g., Phys. Rev. Appl. **11**, 034044 (2019)].

While there have been fewer experiments focussed on studying light propagation in these systems (light interaction in 3D lattices have been studied, e.g., by the Jun Ye group at Boulder), the numerous light propagation experiments on randomly distributed trapped atom clouds demonstrate that there are no significant barriers to perform these experiments also in lattice systems. We believe the relatively small number of theoretical proposals with high-impact objectives for 3D lattice systems has limited experimental activity so far, and we hope that our manuscript will help stimulate experimental activity in this area.

3. Another criticism I have on the paper is that the entire analysis is performed in the low driving limit, which makes the momentum space representation the authors use exact. This approach has been extensively exploited in the literature over the past years, and is known as an efficient way to characterize light propagation through cooperative arrays. However, it would be valuable to include a discussion or preliminary analysis of how higher light intensities and resulting nonlinearities might affect the negative refraction, potentially altering the conclusions drawn under the low-intensity approximation.

One of the advantages of the atomic systems, as compared with artificial resonators, is indeed the ability to study quantum and strongly nonlinear responses. We then thank the referee for stimulating further investigation in this regard. While detailed studies beyond the low light intensity (LLI) limit would merit separate full research projects on their own, we have now carried out a preliminary investigation into these effects in the numerically amenable configuration of Fig. 5(a), in the updated manuscript. We have employed a semi-classical approximation where the quantum correlations between different atoms are neglected (for fixed atomic positions, this represents a mean-field theory). This approximation has been shown to describe the optical responses of the relevant atomic arrays accurately outside the regimes of phase transitions, with many-atom quantum effects representing only small corrections [see Ref. [52] and, e.g., Phys. Rev. A **108**, 013711 (2023)]. Our preliminary study focusses on weak nonlinearities to demonstrate that negative refraction is not exclusively a phenomenon of linearly responding systems and can persist even in the presence of nonlinearities. Numerically, even this preliminary study is non-trivial, as it involves over 3000 radiatively coupled nonlinearly responding atoms. We find that layer excitations are effectively reduced in proportion to their relative population in the resonant eigenmode in the LLI limit. Similarly to the phenomenological models for imperfect filling fractions and position fluctuations, this results in decreased coherent light transmission and beam displacement. In addition,

the nonlinear response results in a coupling to nearby k -space modes other than those excited by the beam in the LLI limit, which in turn leads to spatial broadening of the beam.

The enclosed figure illustrates the modified transmission due to the nonlinear response. We have also included it as a Supplementary Fig. 4 in the modified version of the manuscript.

FIG. 1. Negative refraction of light (a) in the low-light-intensity limit and (b) beyond the low light intensity limit when the atoms respond nonlinearly. The configurations are identical to Fig. 5(a) of the Main Text in the manuscript.

We find that qualitatively the atomic response of the system to nonlinearity is analogous to that of a single planar atomic array. We here assume an incident plane with the wave vector \mathbf{k} , and a phase varying Rabi frequency, $\mathcal{R}_l = \mathcal{R}e^{i\mathbf{k}\cdot\mathbf{r}_l}$, where \mathbf{r}_l denote the atomic positions in the array. A general solution for a single atomic layer can be expressed [Commun. Phys. **3**, 205 (2020)] in terms of the excited level populations $\rho_{ee}^{(l)} = \rho_{ee}$ and the (ground-excited level) coherences $\mathcal{P}^{(l)} = \mathcal{P}e^{i\mathbf{q}\cdot\mathbf{r}_l}$ ($\mathbf{q} = \mathbf{k}$), where ρ_{ee} and \mathcal{P} are uniform functions, with

$$\mathcal{P} = \frac{i\mathcal{R}Z}{i[\Delta - Z\delta_s(\mathbf{q})] - [\gamma - Z\tilde{\gamma}_s(\mathbf{q})]}, \quad (1)$$

where

$$Z = 2\rho_{ee} - 1, \quad (2)$$

and $\delta_s(\mathbf{q})$ and $\gamma + \tilde{\gamma}_s(\mathbf{q})$ are the LLI collective eigenmode line shifts and linewidths, respectively. The excited level population influences through Z both the resonance width and shift (similarly to imperfections in Eq. (12) of the Main Text), as well as suppressing the total excitation.

Whilst we are pleased to strengthen the manuscript with this inclusion, we want to emphasise that consideration of nonlinear responses are by no means necessary for our proposal since the LLI limit can be reached experimentally, as demonstrated by the single atomic array experiments Refs. [22,23], and even the recent light scattering experiments on randomly distributed atoms (arXiv:2409.04148). As also acknowledged by Reviewer 1, it is “an efficient way to characterize light propagation” which enabled our succinct demonstration and explanation of negative refraction.

Response to Reviewers 2 and 3

The manuscript “Negative refraction of light in atomic medium” presents a rigorous numerical analysis of the idea to use an ordered atomic array with sub-wavelength spacing to generate a negative index of refraction of the atomic medium at optical frequencies. The negative refraction angle results from cooperative effects between the atoms for some light-atom detunings and certain ranges of incidence angle. The authors cleverly reduce the complexity of the problem by decomposing the atomic polarization dynamics within a single atomic layer into Bloch waves propagating along in-plane wavevectors. This allows them to simplify the spectral analysis of the coupled-oscillator problem, which leads to a clear picture of dispersion band folding, resulting in the effective negative refractive index. The numerical analysis is further extended to include typical experimental imperfections such as finite temperature of the atoms, disorder in the trap array positions, and missing atoms. The authors conclude that even in the presence of these imperfections, the negative refraction angle should be observable with state-of-the-art experimental techniques. The article is well written and logically structured, and easy to follow. Cited references establish a good context for the presented work, and support the arguments made by the authors. While we cannot directly verify the calculations in the manuscript, the approach taken by the authors is reasonable, and the results appear valid. The topic of a negative index of refraction at optical frequencies is an interesting one, and the proposed approach may be superior to nanofabricated systems. We believe that the manuscript will be of interest to the readership of Nature Communications, and we recommend publication. However, we ask the authors to comment on the following questions.

1. The authors show the negative displacement of an incoming beam for a certain range of input angles. However, the discussion of the negative index of refraction in the introduction is quite general, and at least for the simplest applications one would want the negative index to hold for all (or at least, most) incidence angles. To what extent do the demonstrated results actually generalize to a situation with a negative index of refraction, rather than just some unusual dispersion? Is the

system as calculated useful for applications? The authors criticize in the introduction that systems with photonic crystals “are limited to narrow range of incidence angles and frequencies” – how does the proposed system compare to photonic crystals?

We thank Reviewers 2 and 3 for their time and effort dedicated to refereeing our work, and are pleased to see their positive comments. Indeed, we have shown in Fig. 1 that high-transmission negative refraction is observed in almost a full range of incident angles ($\pi/6 \lesssim \theta \lesssim \pi/2$, implying a broad range of quasimomentum excitations that are not only restricted to some cases of anomalous dispersion) across the transmission band, even without fine-tuning the lattice geometry or optimising parameters. Our conclusions hold equally for beams incident in both the xy and xz planes, demonstrating that negative refraction is present in a 3D angular range. In contrast, current implementations at optical frequencies in photonic crystals are limited to two dimensions, effectively restricting the incident beam to lie in a plane. In 3D crystals, negative refraction has so far only been observed at microwave frequencies, whilst existing 3D structures that may be suitable at optical frequencies commonly feature a spatial period comparable to the wavelength. This results in undesirable higher diffractive orders over an appreciable angular range and limits any applications requiring subwavelength resolution. Moreover, the control range of the effective index magnitude is limited within the refractive indices of the constituent dielectrics, which themselves entail complex fabrication procedures for fully 3D structures. Atoms have no such constraints, as 3D subwavelength atomic arrays operating at optical frequencies are a readily available technology (see Response 2 to Reviewer 1). We have clarified this further in “Introduction”.

Nonetheless, demonstrating advanced applications of negative refraction constitute significant and highly non-trivial follow-up projects on their own, which we plan to undertake in the future. This is clearly outside of the scope of the current study. However, natural media made of atoms offer pristine conditions that are extremely promising and potentially crucial for further development of negative refraction technologies, as atoms do not suffer from the imperfections that affect the artificially manufactured metamaterials, with the further advantage of greatly simplifying the inverse design problem when optimising for optical response. We have elaborated more on this point in Response 2 to Reviewer 4 and in Response 1 to Reviewer 1.

Our work primarily establishes the microscopic principles of negative refraction in atomic media, suggesting that with further optimisation, these media can stimulate the investigation and development of negative refraction technologies.

2. The article offers a well-reasoned numerical study, yet it lacks a tangible and intuitive microscopic

explanation for the observed effects. Can the authors think of an intuitive mechanism behind the phenomenon studied in the article? (A microscopic picture could perhaps be elucidated based on the eigenvectors in the spectral analysis.) Could the authors attempt to explain the effects they observe based on physics arguments, perhaps with some pictures of microscopic polarization profiles? Is such explanation feasible in light of cooperative effects, and if not, why not? An answer to these questions is not a requirement for publication in Nature Communications, but such an explanation would make the manuscript much stronger, and more interesting to a general audience.

In the original submission, we provided an explanation based on the transverse band structure, using the specific momentum-space representation of the infinite-layer analysis. In this description, refraction is understood by analysing the resonances of collective eigenmodes, each associated with a well-defined in-plane quasimomentum. The accompanying eigenvalues define the collective resonance linewidths $v^{(j)}(\mathbf{q}_{\parallel})$ and line shifts $\delta^{(j)}(\mathbf{q}_{\parallel})$ for each band j . The incident light beam excites a wave packet centered around the wavevector \mathbf{q}_{\parallel} , which is phase-matched and resonant with the beam, as illustrated in Figs. 1(c,d).

The refractive index can be qualitatively understood by a simple physical argument: According to standard theory, the group velocity of a wave packet is determined by the gradient of the dispersion band, $\mathbf{v}_g = -\nabla_{\mathbf{q}}\delta^{(j)}(\mathbf{q})$, where the derivative is evaluated with respect to the quasiwavevector \mathbf{q} . The light-induced excitation wave packet, matching the incident light wavevector component \mathbf{k}_{\parallel} , experiences transverse displacement along the y axis due to propagation at the group velocity component in that direction, $v_{g,y} = -\partial\delta^{(j)}(\mathbf{q}_{\parallel})/\partial q_y$, evaluated at $\mathbf{q}_{\parallel} = \mathbf{k}_{\parallel}$. The lifetime of this excitation is given by the inverse collective linewidth of the excited eigenmode $1/v^{(j)}(q_y, q_z = 0)$.

Therefore the transverse beam displacement – and thus the resulting group index of refraction based on Snell’s law – can be approximated by the formula $\tilde{D} \simeq v_{g,y} \times 1/v^{(j)}(q_y, q_z = 0)$, the velocity of the wave packet times the lifetime of the excitation. This description also reveals the intriguing result that refraction can be enhanced by Dicke subradiance due to smaller values of $v^{(j)}(q_y, q_z = 0)$.

It is also interesting to note that resonances in the in-plane polarisation band structure of the medium align with resonances in the observed transmission spectrum. This highlights how the observed transmission can be directly understood as Fano resonances, arising from interferences between broad superradiant eigenmodes and the narrow subradiant modes observed in Fig. 3.

Following the Reviewers’ suggestion, we have expanded the original microscopic description of the

FIG. 2. **Microscopic origin of negative refraction and emergence of macroscopic optical response.** (a) Schematic illustrates the microscopic mechanism of negative refraction. An atomic polarisation wavepacket, excited by the incident beam, propagates along the y -axis over its lifetime $t = 1/v$, where $v = v^{(j)}(q_y, q_z = 0)$ is the collective linewidth, accumulating a transverse displacement of $D \simeq v_{g,y} \times 1/v$. The transverse group velocity component $v_{g,y} = -\partial\delta^{(j)}(q_y, q_z = 0)/\partial q_y$ is derived from the collective line shifts $\delta^{(j)}(q_y, q_z = 0)$ for phase-matched quasimomenta q_y in resonant band j . Green and grey wavepackets represent cases of negative and positive displacement, respectively. For (b) $N_x = 25$ (c) $N_x = 50$, and (d) $N_x = 100$ infinite layers, collective line shifts, in units of the single-atom linewidth γ , are presented for the lattice spacing $a = 0.45\lambda$, where λ is the resonance wavelength. The in-plane quasimomentum, indicative of the incident light's tilting angle, is varied. The collective resonance linewidth is normalised to $\gamma/(N_x - 1)$ on a colour-coded logarithmic scale. This choice highlights the linear dependence of the wavepacket lifetime, and displacement D , on sample thickness $a(N_x - 1)$, as alluded to in (a). Anomalous bright dots correspond to resonances due to array edges in the x -direction.

refractive index and also included a schematic illustration of the effect in Fig. 4 of the manuscript (enclosed below).

3. Could the authors provide a paragraph briefly summarizing the parameter regimes and imperfection thresholds for observing the negative refraction? How does the refraction deteriorate when approaching the thresholds (e.g. in the required atom filling)? Are there some intuitive guidelines for picking the best experimental parameters to observe the phenomenon? Such a summary would be very helpful to experimentalists considering a realization of the proposal.

In the studied examples, we have found generally no sharp threshold for the emergence of negative refraction in terms of atom filling fraction or spatial fluctuations. Neither did we observe a threshold when varying angle. When decreasing the filling, the displacement of the beam changes smoothly, interpolating between the unity filling case of negative refraction and the zero filling case of a trivially unscattered beam. As we discuss in Methods, imperfections, represented by ζ , should be sufficiently small in order to resolve the relevant resonances $\zeta v \gtrsim (1 - \zeta)\gamma$, where the collective (subradiant) and single-atom linewidths are denoted by v and γ , respectively.

We have briefly summarised our findings at the beginning of “Discussion”.

4. The paragraphs explaining the spectral analysis and the methodology of relating its results to the beam displacements (lines 181-207) could appear earlier in the article, e.g. after line 115. This would make it easier to follow the article.

We have carefully considered different options for organising the text. While we agree with Reviewers 2 & 3 that introducing the eigenmodes earlier could be beneficial, the journal’s policy of referring to figures in the order of their appearance complicates this. As we prefer to begin with the plots of the optical responses, instead of more microscopic descriptions, we found maintaining the current order of the text is the simplest and most coherent approach.

5. The colormap in Fig. 1c could be chosen better to highlight the transition from positive to negative displacements. A simple blue-white-red map would work well.

We agree with the Reviewers’ feedback and are grateful for the suggestion. We have switched to a colourmap which highlights the transition while also keeping consistency with the colour palette used in the other figures. (We have also tried other colourmaps such as red-white-blue, and did not find them any clearer.)

Response to Reviewer 4

In the manuscript, the authors theoretically investigated the propagation of light through a 3D atomic array and showed that, under certain conditions, this peculiar atomic medium possesses a negative refractive index. Initially, the authors considered the ideal case of an infinite and defect-free system, but they also moved to the case of finite size, where boundary effects can have important consequences. In both cases, conditions for negative refractive index are found. Interestingly, these results seem to be robust to defects, which means that the phenomenon under

investigation might be observable in the state-of-the-art experiments. I find the manuscript clear, the topic interesting, and the results new and fascinating. Furthermore, the theoretical analysis is deep and precisely carried out. For all these reasons, I recommend publication in Nature Communications. However, I would like to ask the authors to consider the following questions.

1- Even if subwavelength spaced atomic arrays have been realized (see reference 20 and 21), their experimental realization remains challenging. At line 177, the authors claim that it is not important to have equal in-plane and out-of-plane lattice constant for having negative refraction. I wonder if one of these conditions can be relaxed, for example, by having a distance between layers greater than λ (or the opposite configuration). Can similar effects be observed when working with layers of randomly distributed 2D atom clouds?

We thank Reviewer 4 for their time and effort in refereeing our work, and are pleased to see their favourable comments. Although there have been fewer experiments on light transmission through atomic arrays with single-site occupancy, the experimental creation of atomic Mott-insulator states in optical lattice potentials is a well-established technology and nowadays a standard technique in laboratories worldwide. The experimental focus has shifted to more advanced challenges, such as the local control of atoms, the preparation of strongly correlated states that are considerably more difficult to achieve, such as antiferromagnetism in fermionic atoms, and single-site occupation control and resolution imaging. Theoretical proposals exist also for producing subwavelength optical tweezer arrays [see, e.g., Phys. Rev. Appl. **11**, 034044 (2019)]. The numerous light propagation experiments on randomly distributed trapped atom clouds demonstrate that there are no significant barriers to perform more light transmission experiments also in the lattice systems.

We have addressed the experimental feasibility of subwavelength lattices in more detail in Response 2 to Reviewer 1 [for example, a 3D lattice experiment of 800,000 sites in Nature **632**, 267 (2024)].

The effects studied in our manuscript rely on a regular array of atoms, although we also discuss fluctuations from this array and find the effects are robust. We have no evidence that similar behaviour could be observed with entirely random atomic distributions. We have numerically observed negative refraction for a wide range of different subwavelength lattice spacings, including cases of, for example, unequal in-plane ($0.37 \times \lambda$) and out-of-plane ($0.45 \times \lambda$) lattice spacings and have further observed negative refraction with the larger out-of-plane spacing $1.25 \times \lambda > \lambda$, (for the subwavelength in-plane spacing $0.45 \times \lambda$ which ensures that no higher diffraction orders exist), so that the distance between layers may indeed be greater than λ , as the Reviewer alludes to. On

the other hand, as the spacing (along either axis) increases well beyond the wavelength of light, we have observed that the regions of detunings and angles over which negative refraction may be observed tend to decrease, owing to the reduction in strength of dipole-dipole interactions.

2- Although the general interest of the topic treated by the manuscript, it is not clear to me what the prospects are. In the introduction, the authors claim that metamaterials can be useful for the realization of superlensing, for example. Is this a general statement? Or can the specific system investigated be used for this purpose (or any other technological application)?

The main driving force behind the development of artificially manufactured metamaterials has been negative refraction of light, enabling potential applications such as superlensing, allowing focussing and imaging beyond the diffraction limit, as well as transformation optics (see Response 1 to Reviewer 1). Despite the significant interest in negative refraction, intrinsic non-radiative damping and ever-present fabrication imperfections of resonators in metamaterials, compounded by difficulties of inverse design, result in considerable losses at optical frequencies and have permitted only the most basic demonstrations in the nearly 25 years since Pendry's proposal on superlensing. In contrast, natural media made of atoms do not suffer from these imperfections. While formulating advanced applications is a highly non-trivial project on its own, and clearly beyond the scope of the current work, the pristine conditions of atomic media are very promising and potentially crucial for further investigation and development of negative refraction technologies.

3- In the case of a finite system, the authors report their results in Fig 2. According to Fig2a, in steady state, the light intensity is maximized around two layers. This behavior is qualitatively different from that reported in Fig1 for the infinite system, where instead all layers are equally "illuminated". Is there any intuitive explanation for this? Is it related to the number of layers or to the finite size of the sample?

A non-uniform excitation of the layers is generally to be expected, independently of layer number, when exciting subradiant modes. This even happens to subradiant excitations of atoms in a 1D waveguide. In the case of many layers, the intensity profile across the layers is close to that of a resonant phase-matched Bloch wave of the fully infinite 3D system, as, e.g., for standard positive refraction in Supplementary Fig. 1. Although it is more difficult to see, Fig. 1(b) also displays variation across different layers. We finally note that Fig. 2 is obtained for five stacked infinite planar lattices. However, even in the finite-size case (Fig. 4(a) in the original submission, Fig. 5(a) in the modified version), the polarisation density is almost identical, showing that a finite sample

size does not play a significant role in explaining the polarisation profile of Fig. 2.

4- Related to the last point, the authors only considered 5 layers for finite size systems. It is not clear to me how the number of layers matters. According to Fig. 3, apart from the initial 2-3 layers, the displacement D seems to increase linearly with the number of layers, is this correct? I find it difficult to understand Fig 3. Firstly, is it evaluated for a finite or infinite size array? Could it be simpler to plot the displacement D as a function of the number of layers, for example, for two or three values of k_y ?

Reviewer 4 is correct in their observation that, within the transmission band, the displacement D at resonance increases linearly with the number N_x of layers, after an initial transient behaviour for few layers. This general result follows from our findings that (i) the in-plane beam displacement may be approximated by the formula $D \simeq \tilde{D} = v_{g,y}/v$, for in-plane group velocity $v_{g,y}$ and collective linewidth v of resonant phase-matched eigenmodes, as shown in Figs. 3(c,d) (see Response 2 to Reviewers 2 and 3, and now expanded in the modified version of the manuscript); (ii) the corresponding linewidth scales inversely with the number of layers (see Response 1 to Reviewer 1, and now also expanded in the modified version of the manuscript), whilst group velocity tends to a constant. This second point was illustrated in Supplementary Fig. 2 in our original submission. This figure demonstrates how the variation of the number of layers from 25 to 100 effectively maintains the macroscopic refractive response due to the linearly increasing displacement. To address the Reviewers' comments, we have moved this important figure to the main section (Figs. 4(b,c,d)) in the modified version of the manuscript.

Finally, we have updated the manuscript to specify that Fig. 3 is evaluated for arrays infinite in the plane.

5- The simulations for the finite size system were carried out considering a probe beam with a width of 5λ . What exactly do you mean by that? Is it the waist of the beam or its diameter? I'm also wondering about the effect of beam size for a finite system. Indeed, I would expect that a too much focused probe would lead to a reduction of the collective effects, since the light effectively experiences a smaller number of atoms. On the other hand, too large a beam waist would lead to an important effect of the edges of the atomic distribution, again weakening the collective behavior. Is my intuitive picture correct? If so, is there an optimal beam size to enhance the negativity of the refractive index of the material? I think this point could be important in view of a future experimental realization.

The beam waist radius reported here, w , refers to the radius of the beam at the centre, i.e., the amplitude of the Gaussian beam is proportional to $\exp(-\rho^2/w^2)$ at the focal plane, where ρ is the distance from the centre of the beam perpendicular to the propagation direction. We have now changed ‘width’ to ‘waist’ and thank the Reviewer for identifying this.

Indeed, too small a beam waist would result in a large spread of transverse momenta, which could in general excite a range of modes and not result in a well-defined exit beam direction. A larger waist would come closer to approximating the plane wave excitation considered for infinite lattices, but can result in scattering from the edges of the arrays. For optimal results of negative refraction of light, it is then generally beneficial to choose a beam waist on the order of the lattice size. However, negative refraction with a well-defined direction for the transmitted light can still be observed for a narrower beam, provided that collective line shifts and linewidths do not vary appreciably across the distribution of excited eigenmodes, as we have confirmed with numerics.

6- All the results presented here are calculated in the low light limit, where each of the emitters behaves as a classical dipole. What is expected beyond this limit? How much the negativity of the refractive is robust to an increasing driving light? Can the quantum nature of a saturated dipole introduce some interesting properties into the system?

We thank the reviewer for prompting further investigation in this regard. Following this suggestion, we have carried out a preliminary study into the effect of weak nonlinearities beyond the limit of low light intensity (see the enclosed figure). We have included Supplementary Fig. 4 to include these findings and refer to the Response 3 to Reviewer 1 for full details.

7- Finally, I have a question out of curiosity. Most of the effects considered here have been evaluated in the steady state. However, I was wondering if the negativity of the diffraction index could also manifest itself in the time domain, for example by studying the modification of the temporal profile of a short light pulse by the propagation of the medium. Could this be another possible observable to measure the negativity of the refractive index? More generally, are there any related collective effects affecting the dynamical evolution before the establishment of the steady state?

We thank Reviewer 4 for stimulating further insight into the nature of negative refraction. This is an interesting point we plan to address more in the future. As a preliminary investigation, we have numerically integrated the time dynamics for negatively refracting light, assuming that the beam is ‘switched on’ at time $t = 0$ (see the enclosed figure which is included in the modified version of manuscript as Supplementary Fig. 2). The dynamics can be explained using our description based

FIG. 3. Snapshots of the normalised light intensity profile (outside the medium) and smoothed atomic polarisation density within the atomic layers, when the light is instantaneously switched on at $t = 0$. The atomic lattice and beam configuration otherwise identical to Fig. 5(a) of the Main Text.

on collective resonances. In particular, the short time dynamics are dominated by the broad superradiant excitation of uniformly excited layers, resulting in beam reflection. On longer timescales, the subradiant mode begins to populate, establishing Fano resonances due to interferences with the broad superradiant mode. As time evolves, light transmission grows together with the increasing beam displacement, as the subradiant contribution is enhanced. The polarisation density of each layer reaches the steady state, beginning with the left-most layer ($x = 0$) and finishing with the right-most layer ($x = 4a$). This result demonstrates that the system does not need to reach steady state for negative refraction to be observed.

8- Even if the number of papers related to this topic is large, I would ask the authors to consider these papers that I find related to the topic of the manuscript: - Bekenstein, R. et al. "Quantum metasurfaces with atom arrays." *Nat. Phys.* 16, 676–681 (2020). - Bettles, R., et al., "Enhanced optical cross section via collective coupling of atomic dipoles in a 2D array." *Physical review letters* 116.10 (2016): 103602. - Solomons, Y., et al., . "Universal approach for quantum interfaces

with atomic arrays.” PRX Quantum 5.2 (2024): 020329. - Asenjo-Garcia A., et al, Exponential Improvement in Photon Storage Fidelities Using Subradiance and “Selective Radiance” in Atomic Arrays, Physical Review X 7, 031024

The first of these (Bekenstein *et al.*), was already cited in the original submission. We are currently over the number of references that the journal allows. Although the other references highlighted by Reviewer 4 represent very important studies of light interacting with atomic arrays, their relevance to the studied system displaying negative refraction in three dimensions may perhaps be less direct than some of the already cited references (which then might need to be removed). We are happy to include extra references, but would first need to consult the journal editors on how much we would be allowed to exceed the specified limit in the total number of references.

9- A final detail: I’m not sure if it is an issue of the version I have, but the quality of the figures is very poor

We suspect this is a problem with the preview version, as all images are PDF vector graphics, but we will check the images on the final version.

List of Changes

The text modifications in the manuscript are highlighted in the attached latexdiff file. Additionally, we made the following changes in the figures:

1. Updated the colormap for Fig. 1(c) in order to highlight the transition of transmission beam displacement through zero, following the suggestions of Reviewers 2 and 3.
2. Corrected a typo in Figs. 3(a,b), so that the lower limit of the y axis now reads 0.0 (previously 0.55).
3. Corrected the plot in Fig. 5(a), in the updated manuscript, so that the atomic polarisation density is correctly presented (the absolute values had previously not been squared), and now a greater match is observed with Fig. 2.
4. Included a schematic, Fig. 4(a), depicting the microscopic origin of negative refraction, following the suggestion of Reviewers 2 and 3.
5. Moved a compressed version of Supplementary Fig. 2 of the original submission to Figs. 4(b,c,d), following the comment of Reviewers 2 and 3.
6. Included the Supplemental Fig. 4 displaying negative refraction in the presence of nonlinearities, following the suggestions of Reviewer 1 and 4.
7. Included the Supplemental Fig. 2, displaying time dynamics and negative refraction before the steady-state, following the inquiry of Reviewer 4.

Response to Reviewers

We again thank Reviewers for their commitment to refereeing our manuscript. While experiments are not easy, we have nevertheless incorporated several non-ideal features in our analysis, such as atomic position fluctuations, finite array size, and missing atoms in a lattice.